# Isolation-based Spherical Ensemble Representations for Anomaly Detection

## Abstract

Tabular Anomaly detection is a critical task in data mining and management with applications spanning fraud detection, network security, and log monitoring. Despite extensive research, existing unsupervised anomaly detection methods still face fundamental challenges including conflicting distributional assumptions, computational inefficiency, and difficulty handling different anomaly types. To address these problems, we propose ISER (Isolation-based Spherical Ensemble Representations) that extends existing isolation-based methods by using hypersphere radii as a monotonic transformation of local density characteristics while maintaining linear time and constant space complexity. ISER constructs ensemble representations where hypersphere radii encode local sparsity through a monotonic transformation of density: smaller radii correspond to dense regions while larger radii correspond to sparse areas. We introduce a novel similarity-based scoring method that measures pattern consistency by comparing ensemble representations against a theoretical anomaly reference pattern. Additionally, we enhance the performance of Isolation Forest by using ISER and adapting the scoring function to address axis-parallel bias and local anomaly detection limitations. Comprehensive experiments on 22 real-world datasets demonstrate ISER's superior performance over 11 baseline methods.

## 1 Introduction

Anomaly detection is the task of identifying data points that deviate significantly from the majority of observations, with applications in fraud detection, network security, and quality control (Chandola et al., 2009; Liu et al., 2024; Tang et al., 2024; Song et al., 2023). Anomalies can manifest in different forms: global anomalies that are isolated in sparse regions far from any clusters, local anomalies that appear normal globally but deviate within their local neighborhood, and dependency anomalies where individual features appear normal but their correlations are unusual Han et al. (2022). Effective anomaly detection methods must handle these diverse types without prior knowledge of which type is present. Despite extensive research, developing effective unsupervised anomaly detection methods remains challenging due to several fundamental limitations.

Existing methods face a critical trade-off between computational efficiency and handling varying local densities. Distance-based methods struggle with datasets having varying local densities, incorrectly flagging normal points in sparse regions. Density-based methods like Local Outlier Factor (Breunig et al., 2000) address this but require quadratic time complexity, limiting scalability. Deep learning approaches capture complex patterns but require extensive training time and lack interpretability. In addition, tabular data presents unique challenges due to feature diversity and lack of natural interconnections. Features demonstrate substantial variation in scales, distributions, and data types (Shenkar & Wolf, 2022), making it difficult to develop universally effective methods.

Isolation-based methods have shown promise by focusing on the principle that anomalies are few and different, making them easier to isolate through random space partitioning (Liu et al., 2008; 2012; Cao et al., 2025). These achieve linear time complexity but traditional methods like Isolation Forest face limitations in detecting local anomalies and suffer from decision boundary imprecision (Hariri et al., 2019).

Recent isolation-based approaches have attempted to address local anomaly detection but face limitations. iNNE (Bandaragoda et al., 2018) uses hypersphere-based partitioning but loses discrimi-

native power when hyperspheres have similar radii. IDK (Ting et al., 2020; 2021) achieves strong performance through kernel mean embedding but suffers from computational overhead that limits practical scalability.

We propose ISER (Isolation-based Spherical Ensemble Representations), which addresses these challenges through a spherical isolation mechanism using hypersphere radii as a monotonic transformation of local density. Our method constructs ensemble representations where smaller radii indicate denser regions while larger radii correspond to sparser areas, enabling effective local density estimation without expensive computations.

ISER provides two distinct scoring mechanisms: average-based scoring computes anomaly scores based on density-aware coverage across multiple partitions, while similarity-based scoring directly computes similarity with a predefined reference anomaly pattern, avoiding kernel mean embedding computations and significantly reducing training time. The key limitations of iNNE and IDK, and the relationship of ISER to them are analyzed in Appendix K.

This paper makes the following contributions: 1) Inspired by iNNE's ratio-based approach, we introduce ISER that leverages hypersphere radii directly as local density monotonic transformation through ensemble representations. 2) We propose a novel similarity-based scoring mechanism that compares ensemble representations against theoretical anomaly patterns, providing robust evaluation complementary to average-based scoring. 3) Additionally, we propose ISER-IF that extends the proposed ensemble representation framework to significantly improve Isolation Forest performance by addressing its axis-parallel bias and local anomaly detection limitations through modified scoring computation adapted to the transformed feature space. 4) Extensive experiments on 22 real-world datasets demonstrate ISER's superior performance over 11 competitive baseline methods, while maintaining the computational efficiency of *linear* time and *constant* space complexity. All the code are available at `https://anonymous.4open.science/r/ISER-5A7E`.

## 2 PROBLEM FORMULATION

According to the standard definition, an outlier is a data point that differs significantly from other observations in a given dataset (Chandola et al., 2009). Given a dataset $\mathcal{D} = \{x_1, x_2, \ldots, x_n\} \subset \mathbb{R}^d$ consisting of $n$ data points in a $d$-dimensional feature space, anomaly detection aims to identify samples that are markedly different from the others in the given observation set.

**Anomaly Types.** Anomalies can be categorized into three main types: global anomalies that deviate significantly from the entire dataset and reside in sparse regions far from dense clusters; local anomalies that appear normal globally but exhibit unusual behavior within their local neighborhood; and dependency anomalies where individual features appear normal but their correlations deviate from expected joint distributions.

**Unsupervised Anomaly Detection.** It is important to distinguish between different anomaly detection settings. In the one-class setting, only normal samples are available during training, and the goal is to learn a detection model that can identify anomalous samples during testing. This setting assumes access to a clean training set containing only normal instances. In contrast, our work focuses on the unsupervised setting, where the dataset is assumed to contain both normal instances (majority) and anomalous instances (minority), the task is to identify anomalous instances in this dataset. This setting is more challenging as it requires distinguishing outliers without any clean reference of normal behavior, relying on the assumption that anomalies constitute a small fraction of the data and exhibit different characteristics from the majority.

## 3 RELATED WORKS

**Distance-based** methods assume anomalies are isolated from their neighbors. k-NN (Ramaswamy et al., 2000) scores points by their distance to the $k$-th nearest neighbor, but struggles in multi-density datasets where normal points in sparse regions are falsely flagged due to naturally large neighbor distances.

**Density-based** approaches identify anomalies as points in low-density regions. LOF (Breunig et al., 2000) computes local outlier factors by comparing a point's local reachability density with that of

its neighbors, enabling detection of local anomalies. Extensions include COF (Tang et al., 2002), which uses chaining distances for better neighborhood continuity, and LOCI (Papadimitriou et al., 2003), which introduces multi-granularity analysis with automatic scale selection. However, these methods suffer from quadratic complexity and sensitivity to neighborhood size.

**Statistical methods** model normal data distributions and flag low-probability instances. HBOS (Goldstein & Dengel, 2012) assumes feature independence and combines histogram bin heights across dimensions, while ECOD (Li et al., 2022) uses empirical cumulative distributions to estimate tail probabilities without binning, improving robustness.

**Isolation-based** methods exploit the fact that anomalies require fewer random splits to isolate. iForest (Liu et al., 2008) uses randomized trees and shorter average path lengths to score anomalies. Variants like SCiForest and Extended iForest improve splitting criteria, while iNNE (Bandaragoda et al., 2018) uses hyperspheres to capture local density ratios with linear complexity. DIF (Xu et al., 2023) integrates deep feature learning with isolation principles.

**Deep learning** methods capture complex patterns using neural networks. Autoencoders and VAEs (Kingma & Welling, 2013) detect anomalies via reconstruction error, though they may fail when anomalies are reconstructible. DeepSVDD (Ruff et al., 2018) learns a compact latent representation, and LUNAR (Goodge et al., 2022) uses GNNs to model local neighborhood structures. DTE Livernoche et al. (2024) detects anomalies by directly estimating the posterior distribution over diffusion timesteps for input samples. These methods achieve strong performance but require extensive tuning and lack interpretability.

## 4 METHODOLOGY

In this section, we present ISER (Isolation-based Spherical Ensemble Representations) that combines the efficiency of isolation-based approaches with the local density estimation. Our approach consists of three main components: space partitioning, ensemble representation construction, and two anomaly scoring methods.

The core idea of ISER is to partition the feature space using hyperspheres centered at randomly sampled points, where each hypersphere's radius is determined by the distance to each sample point's nearest neighbor within the sampled subset. This partitioning process is repeated multiple times with different random subsets to create an ensemble of independent partitions. In each partitioning, dense regions are more likely to have multiple sampled points, resulting in smaller distances between neighboring sampled points and thus smaller hypersphere radii. Conversely, in sparse regions, fewer points are sampled, leading to larger distances between sampled points and correspondingly larger hypersphere radii. Figure 1 demonstrates the key steps of ISER, the key notations are shown in Table 3.

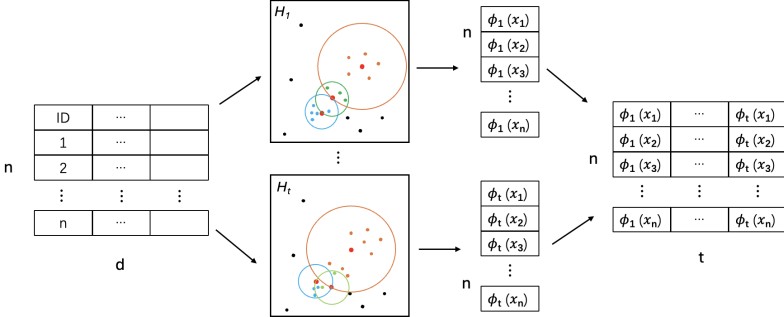

Figure 1: Illustration of ISER ensemble representation construction process. For each partitioning $H_i$, a random subset of $\psi = 3$ points is sampled (shown as red bold centers of each circles in the partitions and input data $d = 2$ in this sample). Each sampled point serves as a hypersphere center with radius determined by the nearest neighbor distance. For each point, $\phi_i$ values from all $t$ partitionings are concatenated to form the final ensemble representation $\Phi(x) \in \mathbb{R}^t$.

## 4.1 SPACE PARTITIONING AND ENSEMBLE REPRESENTATIONS

The proposed ISER employs a hypersphere-based space partitioning mechanism, following the same approach as used in iNNE (Bandaragoda et al., 2018) and IDK (Ting et al., 2020). Given a dataset $D = \{x_1, x_2, \ldots, x_n\} \subset \mathbb{R}^d$, we construct an ensemble of $t$ different partitionings $H_i$, $i = 1, \ldots, t$. For each partitioning $H_i$, we begin by randomly sampling a subset $\mathcal{D}_i \subset D$ of size $\psi$ without replacement, $|\mathcal{D}_i| = \psi$. For stability and diversity, each partitioning is based on a different random subset.

For each point $z \in \mathcal{D}_i$, we build a hypersphere $\theta[z] \in H_i$ centered at $z$. The radius of this hypersphere is determined by the distance between $z$ and its nearest neighbor in $\mathcal{D}_i \setminus \{z\}$:

$$r[z] = \min_{z' \in \mathcal{D}_i \setminus \{z\}} \|z - z'\|. \tag{1}$$

Each partitioning $H_i$ consists of $\psi$ hyperspheres and the region that is not covered by these hyperspheres. Each hypersphere represents a spherical region in the feature space that contains all points within a certain distance from its center. Each hypersphere $\theta[z] \in H_i$ is thus defined by its center $z$ and radius $r[z]$:

$$\theta[z] = \{x \in \mathbb{R}^d : \|x - z\| \leq r[z]\}. \tag{2}$$

This partitioning mechanism enables the hypersphere radii to serve as a monotonic transformation of local density information. Since the radius of each hypersphere is determined by the nearest neighbor distance within the sampled subset, it directly reflects the local density characteristics of the surrounding region. **Small radii indicate that sampled points are closely packed, suggesting high local density, while large radii indicate that sampled points are widely separated, suggesting low local density.** This density-aware encoding is achieved without explicit density computations, allowing the method to capture essential distributional information efficiently while maintaining the simplicity of geometric partitioning.

For any data point $x \in \mathbb{R}^d$, we construct its ensemble representation by evaluating its relationship with each partitioning $H_i$. The key insight is to encode both the coverage information, indicating whether a point falls within any hypersphere, and the density information, captured by the hypersphere radius.

For each partitioning $H_i$ and point $x$, we first identify the nearest hypersphere center:

$$\hat{z}_i(x) = \arg \min_{z \in \mathcal{D}_i} \|x - z\|. \tag{3}$$

For points at boundaries between regions of different densities, a single partitioning assigns the score based on the nearest hypersphere. However, the ensemble mechanism provides robustness: across $t$ independent partitionings, boundary points are sometimes assigned to hyperspheres from the dense region (low $\phi$) and sometimes from the sparse region (high $\phi$), with aggregated scores naturally reflecting their transitional position.

We then compute $\phi_i(x)$, which directly encodes the density characteristics based on hypersphere radius (as opposed to iNNE's local relative measure (Bandaragoda et al., 2018)), for point $x$ in partitioning $H_i$:

$$\phi_i(x) = \begin{cases} 1 - \frac{1}{r[\hat{z}_i(x)]} & \text{if } x \in \theta[\hat{z}_i(x)] \\ 1 & \text{otherwise.} \end{cases} \tag{4}$$

The rationale behind this formulation is as follows. When a point falls inside the nearest hypersphere, we compute $\phi_i(x) = 1 - \frac{1}{r[\hat{z}_i(x)]}$. As established in Appendix C, this transformation preserves density ordering: smaller radii (characteristic of denser regions) yield smaller $\phi$ values, while larger radii (characteristic of sparser regions) yield larger $\phi$ values approaching 1.

When a point $x$ falls outside the nearest hypersphere $\theta[\hat{z}_i(x)]$, we assign $\phi_i(x) = 1$, indicating that the point is not covered by any hyperspheres in this partitioning. This design choice deliberately ignores local density characteristics for points outside hyperspheres, which is crucial for detecting local anomalies. Through the ensemble of multiple independent partitionings, local anomalies will more frequently fall outside hyperspheres compared to normal points, resulting in consistently higher $\phi$ values regardless of their surrounding density characteristics.

The ensemble representation of point $x$ is constructed by concatenating the coverage indicators from all $t$ partitionings:

$$\Phi(x) = [\phi_1(x), \phi_2(x), \ldots, \phi_t(x)] \in \mathbb{R}^t. \tag{5}$$

The theoretical foundation for using hypersphere radii as a monotonic transformation of density is established in Appendix C, where we prove that the expected radius satisfies $\mathbb{E}[r] \propto \rho^{-1/d}$, providing a monotonic relationship between radii and local density.

## 4.2 AVERAGE-BASED SCORING

The average-based scoring method which call as ISER-A provides a direct approach to anomaly scoring by leveraging the average of the ensemble representation. This approach is grounded in the observation that normal points tend to reside in denser regions and thus consistently exhibit lower ensemble values across multiple partitionings, while anomalous points in sparse regions consistently produce higher values.

The average-based anomaly score is computed as the arithmetic mean of all ensemble components, effectively measuring the average degree of isolation across all partitionings:

$$S_{\text{avg}}(x) = \frac{1}{t} \sum_{i=1}^{t} \Phi_i(x). \tag{6}$$

The interpretation of $S_{\text{avg}}(x)$ is intuitive and directly related to local density characteristics. Higher scores indicate that the point consistently falls outside hyperspheres or within hyperspheres with large radii, both scenarios suggesting the point resides in sparse regions typical of anomalous behavior. Lower scores indicate that the point frequently falls within hyperspheres with small radii, suggesting the point resides in dense regions characteristic of normal behavior. The score is naturally bounded, where values close to 1 correspond to maximally isolated points and lower values correspond to points in denser regions.

## 4.3 SIMILARITY-BASED SCORING

While average-based scoring directly reflects the magnitude of ensemble values, it can be sensitive to rare instances where normal points fall outside all hyperspheres due to incomplete spatial coverage in the random sampling process, resulting in extreme values $\phi = 1$ that may skew the average. Therefore, we propose a similarity-based scoring mechanism which call as ISER-S that measures how well a point's ensemble representation aligns with a theoretical anomaly pattern. This approach leverages the insight that anomalous points should exhibit consistent isolation behavior across multiple partitionings, rather than sporadic isolation in only a few partitionings.

The similarity-based approach begins by defining an ideal anomaly reference pattern that represents the theoretical behavior of a perfect anomaly. We define this anomaly reference vector as:

$$\mathbb{K} = [1, 1, \ldots, 1] \in \mathbb{R}^t. \tag{7}$$

This vector represents the theoretical pattern of a point that lies outside all hyperspheres across all partitionings, indicating maximum and consistent isolation in every dimension of the ensemble space. Such a pattern would be exhibited by a point that is consistently far from all local density centers, representing the most extreme form of anomalous behavior.

The similarity-based anomaly score is computed using cosine similarity between the point's ensemble representation and the anomaly reference vector:

$$S_{\text{sim}}(x) = \text{sim}(\Phi(x), \mathbb{K}). \tag{8}$$

The comparison of two scoring approaches are discussed in Appendix J.

## 5 EXPERIMENT

### 5.1 VISUALIZATION ON SYNTHETIC DATASETS WITH DIFFERENT ANOMALY TYPES

All experimental settings are provided in Appendix E. Table 1 presents the performance comparison between different isolation-based methods using the same hypersphere partitioning strategy. For

Table 1: Performance comparison of isolation-based methods using the same hypersphere partitioning strategy on local and dependency anomaly.

| | iNNE | IDK | ISER-A | ISER-S |
|---|---|---|---|---|
| S1 | AUROC = 0.8702
AUPR = 0.8271 | AUROC = 0.8330
AUPR = 0.7742 | AUROC = 0.8445
AUPR = 0.8029 | AUROC = 0.9134
AUPR = 0.8273 |
| S2 | AUROC = 0.9883
AUPR = 0.9450 | AUROC = 0.9698
AUPR = 0.8842 | AUROC = 0.9961
AUPR = 0.9761 | AUROC = 0.9956
AUPR = 0.9723 |

local anomalies, the proposed ISER demonstrates superior detection capability through similarity-based scoring mechanism. While local anomalies may locate close to dense regions, but they fall outside all hyperspheres compared to truly normal points across multiple random partitions. The similarity-based scoring effectively captures these subtle but consistent differences by comparing ensemble representations against the ideal anomaly reference pattern. For dependency anomalies, ISER demonstrates detection capability through its sampling bias toward the dominant distribution patterns. Since random sampling naturally favors the majority distribution (normal feature correlations), the constructed hyperspheres tend to center around regions that conform to typical feature relationships. Points with dependency anomalies, which violate these dominant correlation patterns, are more likely to fall outside the majority-biased hyperspheres across multiple partitions, creating distinctive ensemble signatures that enable detection. For global anomalies, all methods achieve perfect performance, the demonstration is in Appendix F.

The visualization results reveal a crucial distinction in decision boundary characteristics. ISER exhibits significantly more compact and precise decision boundaries that tightly conform to the normal data distribution, while iNNE and IDK show more diffuse, stratified boundaries with smoother gradients extending into sparse regions. This boundary compactness is consistently observed across different anomaly scenarios.

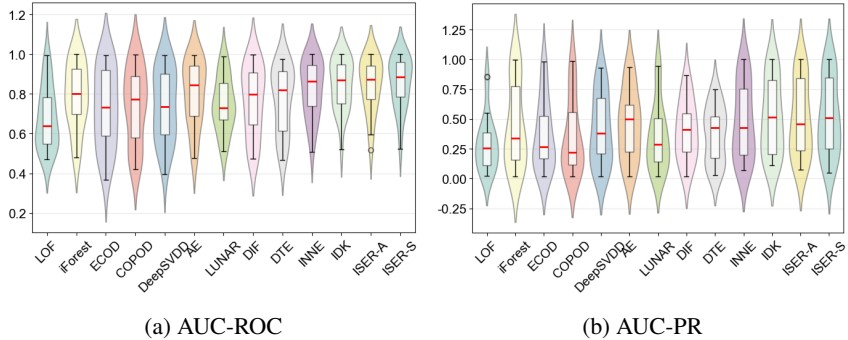

(a) AUC-ROC        (b) AUC-PR

Figure 2: Comparison of all methods performance in terms of AUC-ROC and AUC-PR.

## 5.2 EMPIRICAL EVALUATION ON REAL-WORLD DATASETS

Figure 2 shows the performance comparison using combined box and violin plot visualizations. The boxplots reveal clear performance hierarchies: ISER methods achieve the highest median performance for both AUC-ROC and AUC-PR metrics. ISER-S shows a median AUC-ROC around 0.9

with a compact interquartile range, indicating both superior average performance and consistency across datasets. ISER-A demonstrates similar characteristics with a median around 0.85, significantly outperforming traditional methods such as ECOD with wider interquartile ranges.

The violin plots complement this analysis by revealing the underlying performance distributions. Both ISER methods show distributions concentrated in the upper performance range. In contrast, some baseline methods exhibit wider spreads with substantial portions extending toward poor performance regions. This distribution analysis indicates that ISER methods not only achieve higher average performance but also demonstrate more reliable performance across diverse dataset characteristics.

The AUC-PR in Figure 2b results reveal even more pronounced differences between methods. While AE and IDK show median values around 0.5 similar to ISER-S, ISER-S demonstrates superior consistency with higher first and third quartile values, indicating more reliable high performance across different datasets. Many other baseline methods show medians near 0.25 or lower, creating a substantial performance gap. ISER-S's combination of high median performance and consistently high quartile values highlights its particular strength in precision-recall scenarios. Full results are shown in Appendix H.

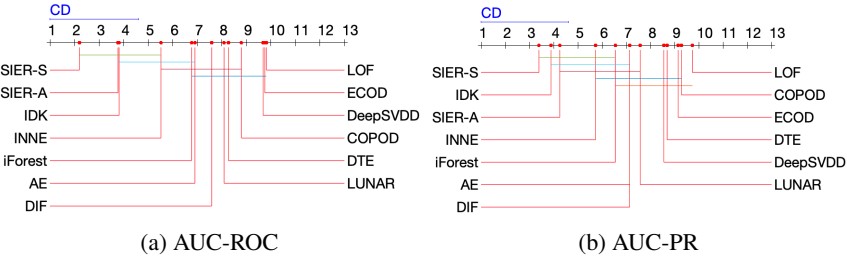

(a) AUC-ROC                (b) AUC-PR

Figure 3: Friedman-Nemenyi test for anomaly detectors at significance level 0.01. If two algorithms are connected by a CD (critical difference) line, there is no significant difference between them.

The critical difference diagrams in Figure 3 provide Friedman-Nemenyi test at significance level 0.01. For AUC-ROC, ISER methods occupy the top two positions. ISER-S is the only method that shows statistical significance over all methods ranked below iForest. For AUC-PR, ISER-S still maintains the first position and remains significantly better than methods ranked below iForest. This establishes the statistical superiority and robustness of the proposed approach across different evaluation criteria. Sensitivity analysis for parameters is detailed in Appendix I.

## 6    SCALABILITY ANALYSIS

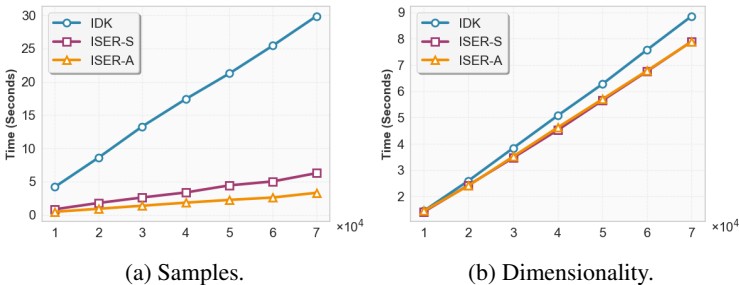

(a) Samples.              (b) Dimensionality.

Figure 4: Runtime comparison. a) The size of dimensionality is 1. b) The sample size is 1000.

**Time complexity:** In the training phase, for each partitioning $H_i$ (with $\psi$ sampled points), nearest neighbor searches are required to compute radii for $\psi$ hyperspheres, each centered at a sampled point. This is done $t$ times to form an ensemble of $t$ partitionings of $\psi$ hyperspheres. The training time complexity is $O(t\psi^2)$.

For evaluating a point $x$, we compute its ensemble representation by finding the nearest hypersphere center in each partitioning (requiring $O(\psi)$ time per partitioning) and evaluating the coverage indicator. The anomaly score computation takes $O(t)$ time. Therefore, the total inference complexity is $O(n\psi t)$. The overall time complexity is **linear** with respect to $n$ because $t$ and $\psi$ are constant hyperparameters and $t\psi \ll n$ for large datasets.

**Space Complexity:** In training phase, ISER only stores the $t$ sets radiis of hyperspheres, resulting in a **constant** $\psi t$ space complexity. In inference phase, ISER computes a $t$-dimensional feature vector to achieve anomaly score for each point.

Figure 4 demonstrates the runtime scalability of ISER methods compared to IDK, the most competitive baseline method, across varying dataset sizes and dimensionalities. The results confirm the linear time complexity of all isolation-based methods, with runtime scaling linearly with both the number of samples and feature dimensions. Notably, ISER methods achieve faster execution times compared to IDK. This performance advantage stems from ISER's similarity-based scoring mechanism, which directly computes similarity with a predefined reference anomaly vector, completely avoiding the computationally expensive kernel mean embedding (KME) calculations required by IDK. ISER-A achieves faster runtime than ISER-S due to simpler arithmetic mean calculation versus cosine similarity computation.

## 7 ISER-BASED ISOLATION FOREST

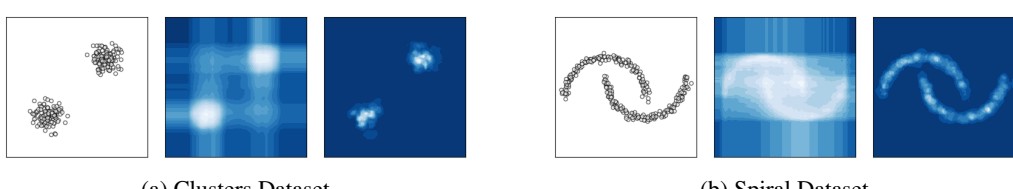

(a) Clusters Dataset          (b) Spiral Dataset

Figure 5: Demonstration of iForest and ISER-IF on different datasets. Left is original data, middle is standard **iForest** and right is the proposed **ISER-IF** in each sub-figure.

iForest has computational efficiency but suffers from inherent limitations that compromise its anomaly detection capability. The axis-parallel splitting criterion restricts branch cuts to hyperplanes aligned with coordinate axes, introducing systematic bias that manifests as rectangular artifacts in anomaly score maps. As demonstrated in Figure 5, in the two-cluster scenario, standard iForest creates artificial low-score regions that extend horizontally and vertically from the cluster centers, forming 'ghost' regions where points receive inappropriately low anomaly scores. The spiral dataset reveals even more dramatic limitations, where the axis-parallel cuts create vertical striping artifacts that obscure the spiral structure and fail to detect local anomalies embedded within the curved geometry.

While methods such as SCiForest (Liu et al., 2010) and Extended Isolation Forest (Hariri et al., 2019) eliminate ghost regions through hyperplane cuts, they still produce scores similar to cluster data in regions between clusters. To address these limitations, we propose leveraging ISER's ensemble representation as a transformed feature space for iForest. The key insight is that ISER's ensemble representation $\Phi(x)$ encodes density-aware spatial information while eliminating axis-parallel bias through its spherical partitioning scheme. By applying iForest to this transformed feature space rather than the original data space, it can inherit ISER's density awareness and local anomaly detection capability.

Once the ensemble representations are constructed using the methodology described in Section 4, we apply iForest to the transformed feature space $\Phi(x)$. However, a critical observation emerges regarding the distribution characteristics in this transformed feature space. Anomalous points, which typically reside in sparse regions of the original space, frequently fall outside hyperspheres across multiple partitionings, resulting in ensemble representations with many components equal to 1. Even when anomalous points fall within hyperspheres, the large radii characteristic of sparse regions yield $\Phi(x) = 1 - \frac{1}{r[\hat{z}_i(x)]} \approx 1$. Consequently, anomalous points are mapped to ensemble representations that exhibit high feature consistency and form tight clusters in the transformed space. In contrast,

normal points residing in dense regions fall within different hyperspheres across various partitionings, and the smaller radii characteristic of dense regions produce $\Phi(x)$ that vary significantly between partitionings, resulting in ensemble representations that exhibit substantial feature diversity and appear more scattered in the transformed feature space.

This creates a fundamental inversion of the iForest assumption. In the ISER feature space, normal points with diverse feature vectors are more easily separated through random cuts, leading to shorter path lengths in the isolation trees. Conversely, anomalous points with highly similar feature vectors form tight clusters that are more difficult to isolate, resulting in longer path lengths. This modification ensures that shorter path lengths in the ISER feature space correspond to lower anomaly scores (indicating normal behavior), while longer path lengths yield higher anomaly scores (indicating anomalous behavior).

To accommodate this inversion, we modify the original iForest anomaly scoring function (Liu et al., 2008) as:

$$S_{\text{ISER-IF}}(x, n) = 1 - 2^{-E(h(\Phi(x)))/c(n)}, \tag{9}$$

where $h(\Phi(x))$ is the path length of $\Phi(x)$ in isolation trees, $E(h(\Phi(x)))$ is the average path length, and $c(n)$ is the normalization constant for $n$ samples.

Figure 5 demonstrates the effectiveness of ISER-IF through visualization comparisons with iForest on two synthetic datasets. The results clearly show that ISER-IF produces significantly more compact and precise decision boundaries that tightly conform to the actual data distribution, completely eliminating the geometric artifacts that characterize traditional isolation methods. In the two-cluster scenario, ISER-IF completely eliminates these artifacts, producing clean, compact boundaries that precisely outline the cluster regions without any axis-aligned extensions. And in spiral dataset, ISER-IF accurately captures the spiral structure, ensuring that the central regions and gaps between spiral arms receive appropriately high anomaly scores while maintaining clean boundaries that follow the natural curvature of the data distribution. Table 2 demonstrates that ISER-IF achieves substantial performance improvements over iForest on datasets where iForest exhibits poor performance.

Table 2: Performance improvement of ISER-IF over iForest with AUC-ROC/AUC-PR.

| Data | iForest | ISER-IF | Improvement(%) |
|---|---|---|---|
| Ionosphere | 0.8530/0.8099 | 0.9314/0.9225 | 9.19/13.90 |
| glass | 0.8016/0.1462 | 0.9014/0.2509 | 12.45/71.56 |
| Fault | 0.5912/0.4277 | 0.7350/0.5331 | 24.32/26.12 |
| Landsat | 0.4945/0.2003 | 0.6272/0.2618 | 26.84/30.70 |
| Pima | 0.6934/0.5304 | 0.7360/0.5655 | 6.14/6.62 |
| satellite | 0.7149/0.6672 | 0.7839/0.7063 | 9.65/5.86 |
| speech | 0.4803/0.0174 | 0.6443/0.0911 | 34.15/423.56 |
| vowels | 0.7647/0.1376 | 0.9526/0.3958 | 24.57/161.48 |
| Waveform | 0.7221/0.0574 | 0.7333/0.0815 | 1.55/41.99 |
| WPBC | 0.5215/0.2266 | 0.5936/0.3148 | 13.83/38.92 |

## 8 CONCLUSION

In this work, we present ISER that integrates spherical ensemble representations with density-aware scoring mechanisms. The proposed approach uses hypersphere radii as a monotonic transformation of local density to enable effective detection of both global and local anomalies while maintaining linear time complexity. The empirical evaluation on 3 synthetic datasets and 22 real-world datasets demonstrates that ISER achieves superior performance compared to 11 representative baseline methods. The ability of ISER to enhance iForest through ensemble representations further validates its practical utility. Future work could explore adaptive parameter selection strategies and investigate the framework's applicability to streaming anomaly detection scenarios.

## ETHIC STATEMENT

This research presents a novel anomaly detection method that operates on numerical tabular data without involving human subjects or sensitive personal information. The ISER algorithm uses standard benchmark datasets from established repositories (PyOD and ODDS) that are publicly available for research purposes. While anomaly detection techniques can be applied in various domains including security and fraud detection, the research focuses on methodological contributions rather than specific applications that might raise ethical concerns. The authors have made efforts to ensure fair comparison with baseline methods and transparent reporting of limitations. We acknowledge our adherence to the ICLR Code of Ethics throughout this research. Use of LLMs is claimed in Appendix L.

## REPRODUCIBILITY STATEMENT

We have made comprehensive efforts to ensure the reproducibility of our results. All experimental details, including hyperparameter settings for both ISER and baseline methods, are provided in Appendix E. The complete algorithm implementation is detailed in Algorithm 1. Datasets information is thoroughly documented in Table 4. Source code for ISER implementation and experimental scripts has been made available at the anonymous link: `https://anonymous.4open.science/r/ISER-5A7E`. Sensitivity analysis for parameters are documented in Appendix I.

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

# A    FULL RELATED WORKS

Existing unsupervised anomaly detection methods can be broadly categorized based on their underlying assumptions and methodologies.

**Distance-based** approaches assume that anomalies are far from their neighbors, while normal points are surrounded by neighbors (Chandola et al., 2009). Representative methods include $k$-nearest neighbor (k-NN) based approaches such as KNN (Ramaswamy et al., 2000), which compute anomaly scores based on the distance to the $k$-th nearest neighbor. However, this approach struggles with datasets exhibiting varying densities, as normal points in sparse regions may be incorrectly classified as anomalies due to their naturally larger neighbor distances.

**Density-based** methods assume that normal data occurs in dense regions while anomalies occur in sparse regions. LOF (Local Outlier Factor) (Breunig et al., 2000) is the most prominent example, computing local density and identifying points with significantly lower local densities as anomalies. LOF introduces the concept of local reachability density, which measures the density of a point relative to its neighbors, and the local outlier factor, which compares a point's density to that of its neighbors.

Related methods extend this concept in various directions. COF (Connectivity-based Outlier Factor) (Tang et al., 2002) improves LOF when a pattern itself has similar neighbourhood density as an outlier by using chaining distances. LOCI (Local Correlation Integral) (Papadimitriou et al., 2003) introduces the concept of multi-granularity deviation factor, providing automatic parameter selection and statistical significance bounds for outlier detection.

While density-based methods excel at detecting local anomalies and handling varying densities, the quadratic time complexity arises from the need to compute neighbor relationships for all points, making them unsuitable for large-scale datasets. Additionally, these methods are highly sensitive to neighborhood parameter choices (such as $k$ in k-nearest neighbors).

**Statistic-based** approaches model the data distribution and identify points with low probability as anomalies. These methods assume that normal data follows a known or learnable probability distribution, and anomalies are points that fall in low-probability regions of this distribution. HBOS (Histogram-based Outlier Score) (Goldstein & Dengel, 2012) assumes feature independence and constructs histograms for each feature dimension. The anomaly score is computed as the product of inverse bin heights across all dimensions, effectively measuring how rare a point's feature combination is.

ECOD (Empirical Cumulative Distribution based Outlier Detection) (Li et al., 2022) addresses some limitations of histogram-based methods by using empirical cumulative distribution functions (ECDF) to estimate feature distributions. ECOD computes tail probabilities for each feature and aggregates them to produce anomaly scores.

The primary limitation of statistical methods lies in their distributional assumptions. Real-world data often exhibits complex, unknown distributions that cannot be adequately captured by standard statistical models.

**Isolation-based** approaches assume that anomalies are few and different, making them easier to isolate than normal points. This principle fundamentally differs from traditional methods that try to profile normal behavior. Isolation Forest (iForest) (Liu et al., 2008) is the seminal work, using random trees to partition the data space and measuring path lengths required for isolation. The algorithm recursively partitions the data space using random hyperplanes, with the intuition that anomalies require fewer partitions to be isolated due to their rarity and distinctiveness. The anomaly score is computed based on the average path length across multiple isolation trees, with shorter paths indicating higher anomaly likelihood.

Other variants include SCiForest (Liu et al., 2010), which uses random hyperplanes instead of axis-parallel splits to address the bias in axis-parallel partitioning, particularly in high-dimensional spaces. Extended Isolation Forest (Hariri et al., 2019) introduces slope-based partitioning to create more balanced and effective isolation trees.

Despite their efficiency and effectiveness, isolation-based methods face challenges in detecting certain types of anomalies, particularly local anomalies embedded within dense regions. To address this

challenge, iNNE (Isolation-based Nearest Neighbor Ensemble) (Bandaragoda et al., 2018) addresses local anomaly detection limitations by using hypersphere-based isolation. iNNE constructs hyperspheres centered at randomly sampled points with radii determined by nearest neighbor distances, naturally incorporating local density ratio while maintaining linear time complexity. More recently, IDK has incorporated kernel-based approaches, using data-dependent kernels to measure distribution similarities for improved anomaly detection performance. DIF (Deep Isolation Forest) (Xu et al., 2023) extracts features with neural networks and then apply standard iForest on it.

**One-class Classification** methods learn a boundary around normal data, treating anomaly detection as a one-class learning problem. One-Class SVM (OCSVM) (Schölkopf et al., 1999) maps data to a high-dimensional space using kernel functions and finds a hyperplane that separates normal data from the origin with maximum margin. The method implicitly assumes that normal data clusters around the origin in the transformed space, while anomalies lie farther from the origin. OCSVM's effectiveness heavily depends on kernel choice and parameter tuning.

SVDD (Support Vector Data Description) (Tax & Duin, 2004) finds a hypersphere that encloses normal data with minimal volume in the feature space. Unlike OCSVM, SVDD directly optimizes for a compact description of normal data without requiring a specific origin. SVDD identifies support vectors that define the hypersphere boundary, and anomaly scores are computed based on distances to this boundary. Both OCSVM and SVDD can handle non-linear decision boundaries through kernel functions but suffer from computational complexity issues with large datasets and sensitivity to hyperparameter selection.

**Deep Learning-based Methods.** Recent advances have led to sophisticated methods based on neural networks that can capture complex non-linear patterns in data. Autoencoders detect anomalies based on reconstruction errors, operating under the assumption that normal patterns can be compressed and reconstructed accurately, while anomalous patterns result in higher reconstruction errors. However, this assumption can be violated when anomalies lie on the learned manifold or when the autoencoder has sufficient capacity to reconstruct anomalies. Variants include Variational Autoencoders (VAE) (Kingma & Welling, 2013), which introduce probabilistic modeling and regularization to improve generalization and provide uncertainty estimates.

DeepSVDD (Ruff et al., 2018) combines deep learning with one-class classification principles, training neural networks to map normal data to a hypersphere with minimal volume in the learned representation space. LUNAR (Goodge et al., 2022) uses graph neural networks to learn trainable representations of local neighborhood information for anomaly detection

While deep learning methods can capture complex patterns and achieve strong performance on high-dimensional data, they typically require extensive computational resources, careful hyperparameter tuning, and large amounts of training data. Their black-box nature also makes interpretation challenging, and they may overfit to specific datasets without proper regularization.

## B  KEY NOTATIONS

Table 3 shows the key symbols and notations in this paper.

## C  THEORETICAL FOUNDATION FOR RADIUS-BASED DENSITY ENCODING

In this appendix, we provide a rigorous theoretical justification for using hypersphere radii as a monotonic transformation of local density. For the theoretical analysis, we define the transformation function $f(r) = 1 - 1/r$. From Equation 4, when point $x$ falls within a hypersphere, $\phi_i(x) = 1 - \frac{1}{r[\hat{z}_i(x)]} = f(r[\hat{z}_i(x)])$. We prove that the expected radius of hyperspheres constructed through our random sampling procedure exhibits a monotonic relationship with the underlying data density, thereby validating the use of $f(r) = 1 - 1/r$ for density-aware anomaly detection.

### C.1  PROBLEM SETUP

Consider a dataset $\mathcal{D} = \{x_1, x_2, \ldots, x_N\} \subset \mathbb{R}^d$ consisting of $N$ data points in a $d$-dimensional feature space. The dataset exhibits spatially varying density characteristics across different regions.

Table 3: Key symbols and notations

| | |
|---|---|
| $x$ | A data point in input space $\mathbb{R}^d$ |
| $D$ | Dataset containing $n$ points in $\mathbb{R}^d$ |
| $d$ | Dimensionality of the feature space |
| $n$ | Number of points in the dataset |
| $t$ | Number of different partitionings |
| $\psi$ | Size of each random subset |
| $H_i$ | The $i$-th partitioning, $i = 1, \ldots, t$ |
| $\mathcal{D}_i$ | Random subset of $D$ for the $i$-th partitioning |
| $z$ | A point in $\mathcal{D}_i$ serving as hypersphere center |
| $\theta[z]$ | Hypersphere centered at point $z$ |
| $r[z]$ | Radius of hypersphere $\theta[z]$ |
| $\hat{z}_i(x)$ | Nearest hypersphere center to point $x$ in $H_i$ |
| $\phi_i(x)$ | Representation value for $x$ in $H_i$ |
| $\Phi(x)$ | Ensemble representation vector of point $x$ |
| $S_{\text{avg}}$ | Average-based scoring |
| $S_{\text{sim}}$ | Similarity-based scoring |
| $\mathbb{K}$ | A all ones vector |
| $\lVert \cdot \rVert$ | Distance Function |

For a region $V$ with volume $|V|$ containing $N_V$ data points, we define the local density as:

$$\rho = \frac{N_V}{|V|}.$$

Our algorithm constructs hypersphere-based partitions by randomly sampling $\psi$ points from $\mathcal{D}$ (without replacement) to serve as hypersphere centers. Each center $z$ determines a hypersphere radius $r[z]$ based on the distance to its nearest neighbor among the sampled points.

## C.2 Main Theoretical Results

**Proposition 1** (Sampling Distribution). *When randomly sampling $\psi$ points from dataset $\mathcal{D}$, the expected number of sampled points falling in region $V$ is:*

$$\mathbb{E}[\psi_V] = \psi \cdot \frac{N_V}{N} = \psi \cdot \frac{\rho|V|}{N},$$

*where $\psi_V$ denotes the random variable representing the number of sampled points in $V$.*

*Proof.* The sampling process follows a hypergeometric distribution. Each point in $V$ has probability $\psi/N$ of being selected (for large $N$). Therefore:

$$\mathbb{E}[\psi_V] = N_V \cdot \frac{\psi}{N} = \psi \cdot \frac{N_V}{N}.$$

Substituting $N_V = \rho|V|$ yields the result. $\qquad\square$

**Corollary 1** (Sampling Density Preservation). *The sampling density in region $V$, defined as $\rho^{sample} = \frac{\psi_V}{|V|}$, satisfies:*

$$\mathbb{E}[\rho^{sample}] = \frac{\psi\rho}{N}.$$

*Consequently, for two regions with densities $\rho_1$ and $\rho_2$:*

$$\frac{\mathbb{E}[\rho_1^{sample}]}{\mathbb{E}[\rho_2^{sample}]} = \frac{\rho_1}{\rho_2}.$$

*Proof.* By definition, the sampling density in region $V$ is:

$$\rho^{\text{sample}} = \frac{\psi_V}{|V|}.$$

Taking expectation on both sides, and noting that $|V|$ is a constant:

$$\mathbb{E}[\rho^{\text{sample}}] = \mathbb{E}\left[\frac{\psi_V}{|V|}\right] = \frac{\mathbb{E}[\psi_V]}{|V|}.$$

From Proposition 1, we have $\mathbb{E}[\psi_V] = \psi \cdot N_V/N$. Substituting:

$$\mathbb{E}[\rho^{\text{sample}}] = \frac{\psi N_V/N}{|V|} = \frac{\psi N_V}{N|V|}.$$

Since the local density is defined as $\rho = N_V/|V|$, we have $N_V = \rho|V|$. Therefore:

$$\mathbb{E}[\rho^{\text{sample}}] = \frac{\psi \cdot \rho|V|}{N|V|} = \frac{\psi\rho}{N}.$$

For the density ratio, applying the above result to two regions:

$$\frac{\mathbb{E}[\rho_1^{\text{sample}}]}{\mathbb{E}[\rho_2^{\text{sample}}]} = \frac{\psi\rho_1/N}{\psi\rho_2/N} = \frac{\rho_1}{\rho_2},$$

where the constants $\psi$ and $N$ cancel out, establishing that sampling preserves relative density ratios. $\square$

**Proposition 2** (Radius-Density Relationship). *For a hypersphere center $z$ in region $V$ with local density $\rho$, the expected nearest neighbor distance among sampled points satisfies:*

$$\mathbb{E}[r] \approx C \cdot \rho^{-1/d},$$

*where $C = K_d^{-1/d} \cdot (\psi/N)^{-1/d}$ is a constant, and $K_d = \pi^{d/2}/\Gamma(d/2+1)$ is the volume constant for $d$-dimensional hyperspheres.*

*Proof.* Consider a sampled point $z$ in region $V$. We model the nearest neighbor distance by analyzing the expected number of other sampled points within a hypersphere of radius $r$ centered at $z$.

**Step 1: Hypersphere volume.** The volume of a $d$-dimensional hypersphere with radius $r$ is given by the standard geometric formula:

$$V_d(r) = K_d \cdot r^d, \quad K_d = \frac{\pi^{d/2}}{\Gamma(d/2+1)},$$

where $\Gamma(\cdot)$ denotes the Gamma function, a generalization of the factorial function. This reduces to familiar formulas: $V_2(r) = \pi r^2$ for circles and $V_3(r) = \frac{4\pi}{3}r^3$ for spheres.

**Step 2: Expected point count.** The expected number of other sampled points within the hypersphere is the product of sampling density and volume:

$$\mathbb{E}[\text{\# points in } V_d(r)] = \rho^{\text{sample}} \cdot K_d r^d.$$

**Step 3: Nearest neighbor condition.** The nearest neighbor distance $r$ is defined as the radius at which we expect exactly one other sampled point within the hypersphere:

$$\rho^{\text{sample}} \cdot K_d r^d = 1.$$

**Step 4: Solving for radius.** Rearranging:

$$r^d = \frac{1}{\rho^{\text{sample}} \cdot K_d}.$$

Taking the $d$-th root:

$$r = \left(\rho^{\text{sample}} \cdot K_d\right)^{-1/d} = K_d^{-1/d} \cdot \left(\rho^{\text{sample}}\right)^{-1/d}.$$

**Step 5: Approximation via concentration.** From Corollary 1, $\mathbb{E}[\rho^{\text{sample}}] = \psi\rho/N$. From Step 4, we have $r = K_d^{-1/d} \cdot (\rho^{\text{sample}})^{-1/d}$. Taking expectation:

$$\mathbb{E}[r] = \mathbb{E}\left[K_d^{-1/d} \cdot (\rho^{\text{sample}})^{-1/d}\right] = K_d^{-1/d} \cdot \mathbb{E}\left[(\rho^{\text{sample}})^{-1/d}\right].$$

For moderate to large $\psi$, the random variable $\rho^{\text{sample}}$ concentrates tightly around its mean. Under this concentration:

$$\mathbb{E}\left[(\rho^{\text{sample}})^{-1/d}\right] \approx \left(\mathbb{E}[\rho^{\text{sample}}]\right)^{-1/d} = \left(\frac{\psi\rho}{N}\right)^{-1/d}.$$

Therefore:

$$\mathbb{E}[r] \approx K_d^{-1/d} \cdot \left(\frac{\psi}{N}\right)^{-1/d} \cdot \rho^{-1/d} = C \cdot \rho^{-1/d},$$

where $C = K_d^{-1/d} \cdot (\psi/N)^{-1/d}$. $\qquad\qquad\square$

**Corollary 2** (Monotonic Relationship). *The transformation $f(r) = 1 - 1/r$ preserves density ordering. Specifically, for two regions with densities $\rho_1 > \rho_2$:*

$$\mathbb{E}[f(r_1)] < \mathbb{E}[f(r_2)],$$

*where $r_1$ and $r_2$ are the corresponding hypersphere radii.*

*Proof.* From Proposition 2, $\rho_1 > \rho_2$ implies:

$$\mathbb{E}[r_1] \approx C \cdot \rho_1^{-1/d} < C \cdot \rho_2^{-1/d} \approx \mathbb{E}[r_2].$$

Given that $\psi$ is moderate to large, both $r_1$ and $r_2$ concentrate around their respective expectations. For strictly increasing $f(r) = 1 - 1/r$ and concentrated distributions, we have:

$$\mathbb{E}[f(r_1)] \approx f(\mathbb{E}[r_1]) < f(\mathbb{E}[r_2]) \approx \mathbb{E}[f(r_2)],$$

where the approximations $\mathbb{E}[f(r)] \approx f(\mathbb{E}[r])$ hold due to the small variance of $r$ under concentration.

Thus, the transformation preserves the inverse ordering of densities. $\qquad\qquad\square$

### C.3 IMPLICATIONS FOR ANOMALY DETECTION

*Remark* 1 (Nonlinearity and Sufficiency). The dependence of the expected radius on density is nonlinear: $\mathbb{E}[r] \propto \rho^{-1/d}$. The scoring function $f(r) = 1 - 1/r$ does not linearize this relationship. Nevertheless, density-based anomaly detection methods require preserving the ordering of densities. By Corollary 2, $f(r)$ is strictly decreasing in $\rho$, which ensures correct ranking of normal versus anomalous instances. Since AUC-based metrics depend solely on score orderings, this monotonicity is sufficient.

*Remark* 2 (Multi-Density Datasets). Real-world datasets typically contain multiple regions with distinct density characteristics. Our theoretical framework naturally handles such scenarios. Consider a dataset with $m$ regions having densities $\rho_1, \rho_2, \ldots, \rho_m$. Proposition 2 guarantees that for any pair of regions $i$ and $j$:

$$\rho_i > \rho_j \implies \mathbb{E}[r_i] < \mathbb{E}[r_j] \implies \mathbb{E}[f(r_i)] < \mathbb{E}[f(r_j)].$$

When combined with ensemble aggregation over multiple independent partitionings, this pairwise ordering property extends to the entire density spectrum, ensuring that sparser (more anomalous) regions consistently receive higher anomaly scores regardless of the absolute density values.

*Remark* 3 (Boundary Points and Ensemble Aggregation). For data points located at boundaries between regions of different densities, a single partitioning may assign scores based on the nearest hypersphere, which could belong to either the dense or sparse region. However, across the ensemble of $t$ independent random partitionings, boundary points exhibit intermediate behavior:

- Some partitionings sample centers predominantly in the dense region, assigning low anomaly scores.

- Other partitionings sample centers in the sparse region, assigning high anomaly scores.

- The ensemble aggregation (via averaging or similarity scoring) produces intermediate anomaly scores that appropriately reflect the transitional nature of boundary points.

This stochastic averaging mechanism provides robustness to local variations in density and reduces sensitivity to individual partitioning artifacts.

### C.4 COVERAGE-BASED ENCODING FOR LOCAL ANOMALIES

While Proposition 2 addresses radius-based encoding for points within hyperspheres, our method assigns $\phi_i(x) = 1$ when point $x$ falls outside all hyperspheres in partitioning $H_i$. This coverage-based encoding is particularly effective for detecting local anomalies.

*Remark* 4 (Local Anomaly Detection Mechanism). Local anomalies are characterized by deviating from their local neighborhood structure while not necessarily being globally isolated. Consider a local anomaly $x^*$ near a dense region with high density $\rho_{\text{dense}}$. Through random sampling:

1. By Corollary 1, the dense region receives a high concentration of sampled centers.

2. These centers form hyperspheres that cover the normal data manifold within the dense region.

3. The local anomaly $x^*$, while spatially proximate, lies off this manifold and is less likely to be covered by hyperspheres centered on normal points.

4. Across $t$ independent partitionings, $x^*$ exhibits higher frequency of non-coverage ($\phi = 1$) compared to normal points in the same spatial vicinity.

This mechanism allows ISER to distinguish local anomalies from nearby normal points, addressing a key limitation of global density-based methods.

In summary, the theoretical analysis establishes that:

1. Random sampling induces a sampling density that preserves the relative ordering of local densities (Proposition 1, Corollary 1)

2. Hypersphere radii exhibit a power-law relationship with local density: $\mathbb{E}[r] \propto \rho^{-1/d}$ (Proposition 2)

3. The transformation $f(r) = 1 - 1/r$ maintains density ordering despite being nonlinear (Corollary 2)

4. This monotonic relationship is sufficient for anomaly detection as a ranking task, where relative ordering rather than absolute density values determines performance

## D WORKFLOW

Algorithm 1 are the pseudo code for ISER.

## E EXPERIMENTAL SETUP

---

**Algorithm 1** ISER

---

**Require:** Dataset $D = \{x_1, x_2, \ldots, x_n\} \subset \mathbb{R}^d$
**Require:** Number of partitionings $t$, sample size $\psi$
**Ensure:** Ensemble representation $\Phi(x) \in \mathbb{R}^t$ for any point $x$
  1: **for** $i = 1$ to $t$ **do**
  2:     Randomly sample $\psi$ points from $\mathcal{D}$ without replacement
  3:     $\mathcal{D}_i = \{z_1, z_2, \ldots, z_\psi\} \subset \mathcal{D}$
  4:     **for** each $z \in \mathcal{D}_i$ **do**
  5:         Compute radius: $r[z] = \min_{z' \in \mathcal{D}_i \setminus \{z\}} \|z - z'\|$
  6:     **end for**
  7: **end for**
  8: **for** any data point $x \in \mathbb{R}^d$ **do**
  9:     **for** $i = 1$ to $t$ **do**
 10:         Find nearest center: $\hat{z}_i(x) = \arg\min_{z \in \mathcal{D}_i} \|x - z\|$
 11:         **if** $x \in \theta[\hat{z}_i(x)]$ **then**
 12:             $\phi_i(x) = 1 - \frac{1}{r[\hat{z}_i(x)]}$
 13:         **else**
 14:             $\phi_i(x) = 1$
 15:         **end if**
 16:     **end for**
 17:     Construct ensemble representation:
         $\Phi(x) = [\phi_1(x), \phi_2(x), \ldots, \phi_t(x)]$
 18: **end for**
 19: **Anomaly Scoring:**
 20: **Average-based:** $S_{\text{avg}}(x) = \frac{1}{t} \sum_{i=1}^{t} \Phi_i(x)$
 21: **Similarity-based:** $S_{\text{sim}}(x) = \text{sim}(\Phi(x), \Vdash)$

---

**Problem Setting.** Following the unsupervised setting, our experiments assume access to the entire dataset containing both normal and anomalous instances during training, without requiring a clean training set of only normal samples. This differs from one-class settings where methods assume access to pure normal data during training. All methods are evaluated on the same contaminated datasets to ensure fair comparison.

**Datasets.** To demonstrate ISER's versatility in detecting different types of anomalies, we construct three 2D synthetic datasets, each containing a specific anomaly type: global, local, and dependency anomalies. The global anomaly dataset contains normal points forming a circular dense cluster with anomalies distributed in the surrounding sparse regions. The local anomaly dataset features normal points arranged in a spiral pattern with anomalies embedded within or near the spiral structure. The dependency anomaly dataset consists of normal points following one directional pattern while anomalies follow a different directional dependency pattern, demonstrating cases where individual feature values may appear normal but their correlations deviate from the expected relationship.

We also conduct experiments on 22 real-world benchmark datasets from PyOD (Zhao et al., 2019) and ODDS (Rayana", 2016) repositories. The datasets exhibit diverse characteristics in terms of size, dimensionality and contamination rates, as detailed in Table 4. They are from the different domains including healthcare, oryctognosy, astronautics, etc.

**Baseline Methods.** We compare ISER against 11 representative methods: LOF (Breunig et al., 2000), iForest (Liu et al., 2008), ECOD (Li et al., 2022), COPOD (Li et al., 2020), DeepSVDD (Ruff et al., 2018), AutoEncoder, LUNAR (Goodge et al., 2022), DIF (Xu et al., 2023), DTE (Livernoche et al., 2024), iNNE (Bandaragoda et al., 2018), and IDK (Ting et al., 2021). We evaluate both proposed ISER-A and ISER-S.

**Hyperparameter settings.** The hyperparameter of nearest neighbors for LOF and LUNAR is searched in $\{5, 10, 20, 40\}$. For iForest, iNNE, IDK and proposed methods, $\psi \in \{2^m, m = 1, 2, \ldots, 8\}$ and with default $t = 200$. For Autoencoder, DIF and DeepSVDD, the hyperparameter of hidden neuron is searched in $\{[128, 64], [64, 32], [32, 16]\}$. ECOD and COPOD have no parameters.

**Evaluation Metrics.** We use Area Under the ROC Curve (AUC-ROC) and Area Under Receiver Operating Characteristic (AUC-PR) as evaluation metrics. Each experiment is repeated 5 times with average results reported to mitigate randomness.

Table 4: Statistical information of datasets

| Dataset | #Samples | #Features | #Ano. | %Ano. |
|---|---|---|---|---|
| Annthyroid | 7200 | 6 | 534 | 7.42 |
| backdoor | 95329 | 196 | 2329 | 2.44 |
| breastw | 683 | 9 | 239 | 34.99 |
| celeba | 202599 | 39 | 4547 | 2.24 |
| fault | 1941 | 500 | 18568 | 34.67 |
| glass | 214 | 7 | 9 | 4.21 |
| Ionosphere | 351 | 32 | 126 | 35.90 |
| Landsat | 6435 | 36 | 1333 | 20.71 |
| Lymphography | 148 | 18 | 6 | 4.05 |
| mnist | 7603 | 100 | 700 | 9.21 |
| musk | 3062 | 166 | 97 | 3.17 |
| PageBlocks | 5393 | 10 | 510 | 9.46 |
| pendigits | 6870 | 16 | 156 | 2.27 |
| Pima | 768 | 8 | 268 | 34.90 |
| satellite | 6435 | 36 | 2036 | 31.64 |
| satimage-2 | 5803 | 36 | 71 | 1.22 |
| shuttle | 49094 | 9 | 3511 | 7.15 |
| skin | 245057 | 3 | 50859 | 20.75 |
| speech | 3686 | 400 | 61 | 1.65 |
| vowels | 1456 | 12 | 50 | 3.43 |
| Waveform | 3443 | 21 | 100 | 2.90 |
| WPBC | 198 | 33 | 47 | 23.74 |

## F GLOBAL ANOMALY DETECTION

For global anomalies, all methods achieve perfect performance, confirming that isolation-based approaches can effectively detect globally isolated points,which is shown in Table 5.

Table 5: Performance comparison of isolation-based methods using the same hypersphere partitioning strategy on global anomaly.

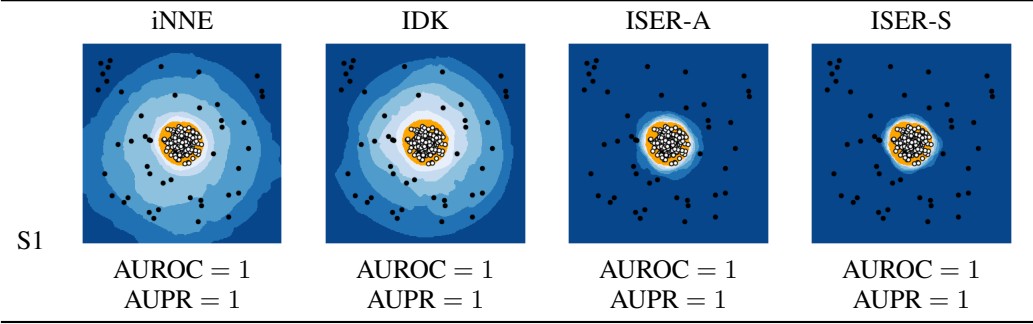

## G DEMONSTRATION OF ISER MAPPING

Figure 6 demonstrates the feature space transformation. In the original space (left), anomalies are scattered while normal points cluster. In $\phi$-space (right), this inverts: anomalies cluster tightly, normal points scatter. Anomalies in sparse regions either fall outside hyperspheres ($\phi = 1$) or within large-radius hyperspheres ($\phi \approx 1$). Both yield high $\phi$ values across partitionings, producing concentrated $\Phi$-representations. Normal points in dense regions fall within hyperspheres of diverse small radii, producing varied $\phi$ values and scattered $\Phi$-representations. This inversion justifies the modified scoring in Eq. (9): in $\phi$-space, the anomaly cluster becomes "hard to isolate" (longer paths in iForest applied to $\phi$), necessitating the inverted scoring rule to correctly identify them as anomalies.

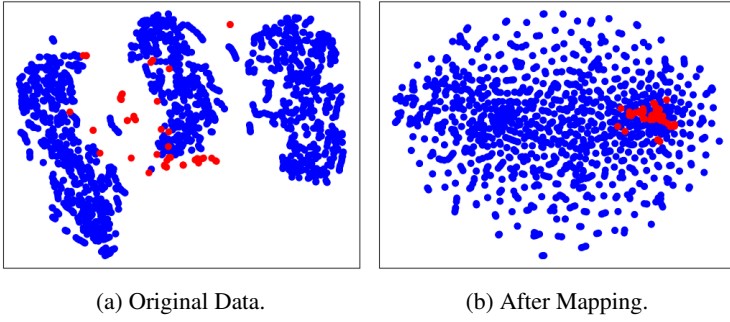

(a) Original Data.    (b) After Mapping.

Figure 6: Demonstration of ISER Mapping for vowel dataset.

# H    FULL COMPARISON RESULTS WITH BASELINE METHODS

Table 6 and 7 present detailed results of all baseline methods and proposed methods across 22 datasets in terms of AUC-ROC and AUC-PR.

Table 6: Results between baselines and proposed methods in terms of AUC-ROC. The best performance are in bold.

| | LOF | iForest | ECOD | COPOD | D.SVDD | AE | LUNAR | DIF | DTE | INNE | IDK | **ISER-A** | **ISER-S** |
|---|---|---|---|---|---|---|---|---|---|---|---|---|---|
| Annthyroid | 0.7362 | 0.8608 | 0.7887 | 0.7760 | 0.6964 | 0.7798 | 0.7367 | 0.6758 | **0.9636** | 0.7455 | 0.7231 | 0.8270 | 0.8030 |
| backdoor | 0.7782 | 0.7516 | 0.8462 | 0.7893 | 0.9026 | 0.9164 | 0.6036 | **0.9189** | 0.8752 | 0.9044 | 0.9044 | 0.9120 | 0.9162 |
| breastw | 0.4691 | **0.9953** | 0.9914 | 0.9944 | 0.9338 | 0.9742 | 0.9780 | 0.7989 | 0.8907 | 0.9822 | 0.9941 | 0.9853 | 0.9859 |
| celeba | 0.4975 | 0.7049 | 0.7572 | 0.7508 | 0.7350 | 0.7374 | 0.5771 | 0.4993 | 0.8122 | 0.8059 | 0.8315 | 0.8088 | **0.8376** |
| fault | 0.5973 | 0.5912 | 0.4687 | 0.4553 | 0.5048 | 0.6898 | 0.7253 | 0.7004 | 0.5895 | 0.6174 | 0.7023 | 0.7210 | **0.7314** |
| glass | 0.8022 | 0.8016 | 0.7046 | 0.7550 | 0.6759 | 0.8439 | 0.8525 | 0.8546 | 0.8639 | 0.7954 | 0.8046 | 0.8730 | **0.8805** |
| Ionosphere | 0.8934 | 0.8530 | 0.7284 | 0.7895 | 0.7850 | **0.9459** | 0.9281 | 0.8975 | 0.9114 | 0.8899 | 0.8883 | 0.9179 | 0.9359 |
| Landsat | 0.5466 | 0.4945 | 0.3678 | 0.4215 | 0.3960 | 0.5168 | 0.6070 | 0.5650 | 0.5442 | 0.6139 | **0.6988** | 0.5963 | 0.6316 |
| Lymphography | 0.9953 | **0.9995** | 0.9953 | 0.9965 | 0.9932 | 0.9930 | 0.9899 | 0.9453 | 0.8341 | 0.9967 | 0.9981 | 0.9969 | 0.9965 |
| mnist | 0.7380 | 0.8132 | 0.7463 | 0.7739 | 0.8428 | 0.8570 | 0.8079 | 0.8825 | 0.8189 | 0.8627 | 0.8708 | 0.8688 | **0.8849** |
| musk | 0.6357 | **1.0000** | 0.9559 | 0.9463 | 0.9727 | 0.9780 | 0.8543 | 0.9894 | 0.9647 | **1.0000** | **1.0000** | **1.0000** | **1.0000** |
| PageBlocks | 0.7861 | 0.8981 | 0.9139 | 0.8754 | 0.8988 | 0.9146 | 0.7834 | 0.8822 | 0.9239 | **0.9494** | 0.9449 | 0.9280 | 0.9212 |
| pendigits | 0.5240 | 0.9517 | 0.9274 | 0.9048 | 0.8581 | 0.8661 | 0.7139 | 0.9560 | 0.7133 | 0.9140 | 0.9340 | 0.9358 | **0.9619** |
| Pima | 0.6399 | 0.6934 | 0.5944 | 0.6540 | 0.6781 | 0.6809 | 0.7053 | 0.6306 | 0.6237 | 0.7010 | 0.7306 | 0.7118 | **0.7356** |
| satellite | 0.5601 | 0.7149 | 0.5830 | 0.6335 | 0.6049 | 0.6847 | 0.6672 | 0.7405 | 0.7105 | 0.7345 | 0.7727 | 0.7658 | **0.7861** |
| satimage-2 | 0.5885 | 0.9945 | 0.9649 | 0.9745 | 0.9619 | 0.9946 | 0.8819 | 0.9971 | 0.9456 | 0.9976 | 0.9983 | 0.9981 | **0.9987** |
| shuttle | 0.5289 | **0.9972** | 0.9929 | 0.9945 | 0.9879 | 0.9948 | 0.6752 | 0.9751 | 0.9762 | 0.9923 | 0.9916 | 0.9928 | 0.9934 |
| skin | 0.5532 | 0.6808 | 0.4888 | 0.4711 | 0.5603 | 0.7054 | 0.7157 | 0.6598 | 0.7405 | 0.7492 | 0.7833 | 0.7779 | **0.7889** |
| speech | 0.6520 | 0.4803 | 0.4697 | 0.4911 | 0.5286 | 0.4760 | 0.5114 | 0.4934 | 0.4946 | 0.6462 | 0.6892 | 0.6437 | **0.6952** |
| vowels | 0.9434 | 0.7647 | 0.5929 | 0.4958 | 0.5884 | 0.9378 | 0.9336 | 0.8268 | 0.9142 | 0.9415 | 0.9528 | 0.9468 | **0.9588** |
| Waveform | 0.7317 | 0.7221 | 0.6035 | 0.7339 | 0.6230 | 0.6506 | 0.7467 | 0.7447 | 0.6020 | 0.7758 | **0.8163** | 0.7901 | 0.7844 |
| WPBC | 0.5201 | 0.5215 | 0.4813 | 0.5233 | 0.5159 | 0.5125 | 0.5204 | 0.4743 | 0.4679 | 0.5080 | 0.5201 | 0.5173 | **0.5240** |
| Avg. | 0.6690 | 0.7857 | 0.7256 | 0.7364 | 0.7384 | 0.8023 | 0.7507 | 0.7776 | 0.7809 | 0.8238 | 0.8432 | 0.8416 | **0.8524** |

# I    SENSITIVITY ANALYSIS

We analyze ISER's sensitivity to its two key parameters: sample size $\psi$ and number of partitionings $t$.

Table 7: Results between baselines and proposed methods in terms of AUC-PR. The best performance are in bold.

| | LOF | iForest | ECOD | COPOD | D.SVDD | AE | LUNAR | DIF | DTE | INNE | IDK | **ISER-A** | **ISER-S** |
|---|---|---|---|---|---|---|---|---|---|---|---|---|---|
| Annthyroid | 0.1812 | 0.3969 | 0.2679 | 0.1724 | 0.2190 | 0.2677 | 0.1880 | 0.2257 | **0.6562** | 0.1880 | 0.1929 | 0.2353 | 0.2311 |
| backdoor | 0.2588 | 0.0492 | 0.0920 | 0.0683 | 0.5198 | 0.5434 | 0.0703 | 0.3985 | 0.5100 | **0.6118** | 0.5913 | 0.5983 | 0.5820 |
| breastw | 0.3093 | **0.9910** | 0.9839 | 0.9886 | 0.9308 | 0.9293 | 0.9432 | 0.5325 | 0.7061 | 0.9568 | 0.9888 | 0.9625 | 0.9627 |
| celeba | 0.0232 | 0.0641 | 0.0967 | 0.0942 | 0.0761 | 0.0613 | 0.0279 | 0.0270 | 0.0987 | 0.1061 | **0.1226** | 0.1058 | 0.0944 |
| fault | 0.4007 | 0.4277 | 0.3247 | 0.3115 | 0.3673 | 0.5171 | 0.5482 | **0.5518** | 0.4410 | 0.4533 | 0.5163 | 0.5251 | 0.5353 |
| glass | 0.1098 | 0.1462 | 0.1771 | 0.0890 | 0.0980 | 0.2244 | 0.1532 | 0.2334 | 0.1385 | 0.1022 | 0.1672 | 0.2439 | **0.2456** |
| Ionosphere | 0.8512 | 0.8099 | 0.6444 | 0.6681 | 0.7112 | **0.9335** | 0.9244 | 0.8684 | 0.7461 | 0.8809 | 0.8821 | 0.9193 | 0.9262 |
| Landsat | 0.2509 | 0.2003 | 0.1632 | 0.1755 | 0.2049 | 0.2178 | 0.2744 | 0.2596 | 0.1790 | 0.2691 | **0.3256** | 0.2493 | 0.2577 |
| Lymphography | 0.8575 | **0.9897** | 0.8873 | 0.9062 | 0.8168 | 0.8243 | 0.8971 | 0.5011 | 0.4147 | 0.9121 | 0.9559 | 0.9175 | 0.9039 |
| mnist | 0.3052 | 0.2829 | 0.1773 | 0.2129 | 0.3925 | 0.4029 | 0.3615 | **0.4327** | 0.3777 | 0.3996 | 0.3975 | 0.4058 | 0.4165 |
| musk | 0.1962 | 0.9994 | 0.4891 | 0.3381 | 0.7580 | 0.6328 | 0.3640 | 0.7998 | 0.5052 | **1.0000** | **1.0000** | **1.0000** | **1.0000** |
| PageBlocks | 0.3779 | 0.4683 | 0.5195 | 0.3704 | 0.4948 | 0.5060 | 0.3926 | 0.5163 | 0.5263 | **0.6410** | 0.6207 | 0.5883 | 0.5677 |
| pendigits | 0.0464 | 0.2642 | 0.2604 | 0.1795 | 0.2146 | 0.0976 | 0.0504 | **0.2852** | 0.0423 | 0.1573 | 0.1964 | 0.1686 | 0.2828 |
| Pima | 0.4379 | **0.5304** | 0.4618 | 0.5223 | 0.5119 | 0.4966 | 0.5067 | 0.4216 | 0.4409 | 0.5053 | 0.5292 | 0.5151 | 0.5301 |
| satellite | 0.3888 | 0.6672 | 0.5260 | 0.5705 | 0.5602 | 0.5719 | 0.4939 | 0.6712 | 0.5787 | 0.6155 | 0.6536 | 0.6536 | **0.6796** |
| satimage-2 | 0.0628 | 0.9359 | 0.6579 | 0.7926 | 0.7728 | 0.7245 | 0.1032 | 0.7671 | 0.1690 | 0.7878 | 0.8804 | 0.8974 | **0.9702** |
| shuttle | 0.1225 | **0.9781** | 0.9040 | 0.9608 | 0.8953 | 0.8882 | 0.1797 | 0.6598 | 0.6826 | 0.9002 | 0.8960 | 0.9047 | 0.9076 |
| skin | **0.5532** | 0.2606 | 0.1820 | 0.1782 | 0.2182 | 0.2768 | 0.2967 | 0.2500 | 0.1949 | 0.3112 | 0.3492 | 0.3360 | 0.3622 |
| speech | 0.0611 | 0.0174 | 0.0176 | 0.0177 | 0.0194 | 0.0170 | 0.0200 | 0.0203 | 0.0262 | 0.0973 | **0.1091** | 0.0939 | 0.0476 |
| vowels | 0.3236 | 0.1376 | 0.0740 | 0.0325 | 0.0482 | **0.5149** | 0.5221 | 0.1699 | 0.4360 | 0.3380 | 0.5098 | 0.3871 | 0.4881 |
| Waveform | 0.0891 | 0.0574 | 0.0391 | 0.0544 | 0.0505 | 0.0542 | 0.1363 | 0.0676 | 0.0600 | 0.0674 | **0.1400** | 0.0736 | 0.0923 |
| WPBC | 0.2247 | 0.2266 | 0.2108 | 0.2292 | **0.2450** | 0.2306 | 0.2295 | 0.2128 | 0.1775 | 0.2225 | 0.2255 | 0.2253 | 0.2273 |
| Avg. | 0.2924 | 0.4500 | 0.3708 | 0.3606 | 0.4148 | 0.4515 | 0.3492 | 0.4033 | 0.3685 | 0.4783 | 0.5114 | 0.5009 | **0.5141** |

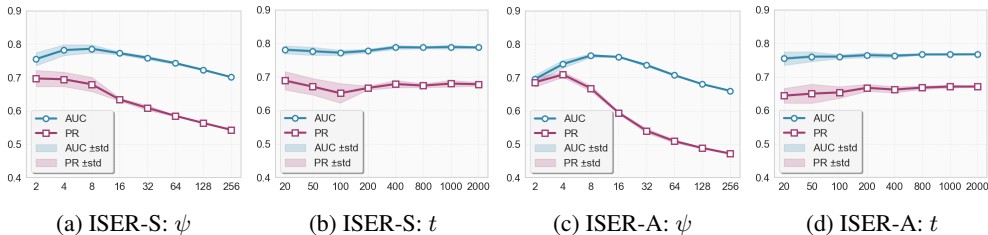

| (a) ISER-S: $\psi$ | (b) ISER-S: $t$ | (c) ISER-A: $\psi$ | (d) ISER-A: $t$ |
|---|---|---|---|

Figure 7: Sensitivity analysis for ISER-S and ISER-A in terms for of AUC-ROC and AUC-PR.

Figure 7a and 7b show that both ISER methods are sensitive to the sample size $\psi$. Performance exhibits a clear peak at moderate values of $\psi$, then gradually declines as $\psi$ increases further. When $\psi$ is too small, there are insufficient hyperspheres to effectively distinguish between normal and anomalous regions, limiting the method's discriminative ability. As $\psi$ increases beyond the optimal point, performance degrades due to excessive hypersphere partitions that create overly fine-grained divisions, leading to overfitting to the training data distribution.

Figure 7c and 7d demonstrate that ISER methods are robust to the number of partitions $t$. Performance remains consistently stable across different values of $t$. This stability occurs because ISER uses random sampling to construct multiple isolators, and averaging across more random samples reduces variance in the final anomaly scores according to the law of large numbers. The performance quickly plateaus, indicating that moderate ensemble sizes are sufficient for stable detection while avoiding unnecessary computational overhead.

## J COMPARISON OF SCORING METHODS

The two scoring methods differ in how they interpret the ensemble representation values. To illustrate these differences, we analyze specific examples with computed scores.

Consider a point with ensemble values [0.9, 0.9, 0.8, 0.9, 0.1] across 5 partitions. The average-based score is 0.79, while the similarity-based score (cosine similarity with reference vector [1,1,1,1,1]) is 0.92. The average-based method is penalized by the single low value (0.1), whereas the similarity-based method emphasizes the predominant pattern of high values, interpreting this as consistent anomalous behavior with one exception due to sampling variation.

For comparison, consider a point with values [0.2, 0.3, 0.3, 0.2, 0.3]. The average-based score is 0.26, appropriately reflecting intermediate anomaly levels based on moderate isolation. The

similarity-based score is 0.98, as the consistent pattern closely matches the reference vector despite lower individual values. In this case, average-based scoring better captures the nuanced density-based anomaly degree, while similarity-based scoring may overestimate the anomaly level.

These examples demonstrate complementary behaviors: average-based scoring is sensitive to the magnitude of individual partition values and thus better reflects varying degrees of local density deviation. Similarity-based scoring emphasizes pattern consistency across partitions, making it robust to occasional normal behavior in individual partitions.

## K   RELATIONSHIP TO ISOLATION-BASED METHODS

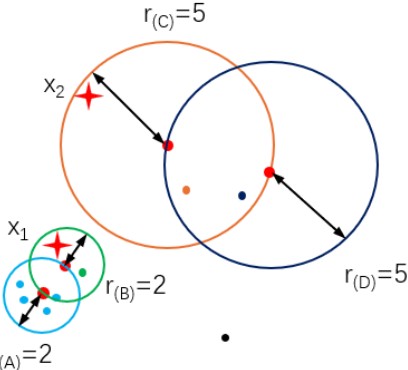

Figure 8: Comparison of scoring methods between iNNE and ISER methods using one same hypersphere partitioning. The radius of hypersphere A and B is 2, and the radius of hypersphere C and D is 5.

While ISER shares the same hypersphere-based space partitioning strategy with iNNE and IDK, the fundamental difference lies in how ensemble representations are constructed and anomaly scores are computed.

**iNNE** computes anomaly scores using local measure rather than global measure, which is a measure of isolation of hypersphere containing $x$ relative to its neighborhood. The anomaly score is defined as $1 - r[\theta[\hat{z}_i(x)]]/r[\eta[\hat{z}_i(x)]]$, where $r[\theta[\hat{z}_i(x)]]$ is the radius of the hypersphere containing point x, and $r[\eta[\hat{z}_i(x)]]$ is the radius of $\theta[\hat{z}_i(x)]$'s nearest neighbor. This local measure approach suffers from a critical limitation: when the containing hypersphere and nearest hypersphere have similar radii, the method loses discriminative power regardless of the actual density characteristics.

In Figure 8, consider the illustration where point $x_1$ resides in a dense region with hypersphere radius $r(A) = 2$, while $x_2$ is located in a sparse region with hypersphere radius $r(C) = 5$. Using iNNE's formulation, both $x_1$ and $x_2$ receive identical scores of 0 (computed as 1-2/2=0 and 1-5/5=0, respectively), despite $x_2$ being in a significantly sparse region and thus more anomalous. Although multiple partitionings can partially alleviate this issue through averaging, the fundamental problem persists and degrades overall performance.

In contrast, ISER directly uses the inverse radius as a density estimation through $1 - \frac{1}{r[\hat{z}_i(x)]}$ when $x$ falls within a hypersphere, which is same to the global measure mentioned by iNNE. This formulation captures absolute density information rather than relative comparisons, yielding $x_1$ score of 0.5 (1-1/2) and $x_2$ score of 0.8 (1-1/5), which correctly reflects that $x_2$ is more anomalous due to its location in a sparse region. This density-aware encoding provides more nuanced anomaly assessment and explains ISER's superior boundary precision observed in synthetic experiments.

**IDK** employs a fundamentally different encoding strategy where each point is represented as a binary vector indicating hypersphere membership. In Figure 8,for one partitioning, if a point falls into hypersphere B, its feature vector becomes [0,1,0,0], while falling into hypersphere C yields [0,0,1,0]. IDK computes the kernel mean embedding (KME) as the average feature vector across

all points, representing the overall data distribution. The anomaly score is then calculated as the similarity (dot product) between each point's feature vector and KME more similar to the overall distribution are considered more normal.

The core issue arises when hyperspheres in different density regions contain the same number of points. In Figure 8,consider the illustration where hypersphere B and C both contain 3 points. This results in a KME vector is [0.36, 0.21, 0.21, 0.14] where the second and third components are identical due to equal point counts. Consequently, point $x_1$ and $x_2$ receive identical anomaly scores despite $x_2$ being located in a significantly sparser region and thus should be more anomalous. This point-counting approach fails to capture the density characteristics that distinguish dense and sparse regions.

In contrast, ISER's radius-based density estimation remains discriminative in such cases, as it directly captures the geometric properties of local density through hypersphere radii rather than relying on point counts. Even when hyperspheres contain equal numbers of points, ISER assigns different scores based on the radii ($x_1$: 1-1/2=0.5, $x_2$: 1-1/5=0.8), correctly reflecting that $x_2$ is more anomalous due to its location in a geometrically sparser region. This fundamental difference in density estimation explains why ISER achieves more consistent performance across diverse data distributions.

To summarize the comparison between iNNE, IDK and ISER:

**iNNE Failure Mode:** When two hyperspheres have similar radii ($r[z_1] \approx r[z_2]$), iNNE's ratio-based score $1 - r[z_1]/r[\eta[z_1]]$ approaches zero regardless of absolute density, losing discriminative power even when the underlying densities differ significantly.

**IDK Failure Mode:** IDK's point-count embedding assigns identical representations to hyperspheres containing equal numbers of points, regardless of hypersphere volume. This causes regions with different densities but equal point counts to be treated as equivalent.

**ISER Advantage:** ISER's absolute encoding $\phi = 1 - 1/r$ remains discriminative as it directly reflects density through $r \propto \rho^{-1/d}$ (Appendix C), avoiding both failure modes while maintaining linear complexity.

# L    USE OF LLMS

This research was conducted without the use of Large Language Models (LLMs) for generating core content, experimental results, or theoretical contributions. The mathematical formulations, algorithm design, experimental methodology, and analytical conclusions presented in this paper are the original work of the authors. While standard writing assistance tools may have been used for grammar checking and language polishing during manuscript preparation, no LLMs were employed for generating research ideas, developing the ISER methodology, implementing algorithms, analyzing experimental data, or drawing scientific conclusions. All code development, experimental design, and result interpretation were performed independently by the research team without AI assistance.

