# OpenReview forum: "Isolation-based Spherical Ensemble Representations for Anomaly Detection"
_ICLR.cc/2026/Conference — Submitted to ICLR 2026_

### Official Review · Reviewer_Kd7K · 2025-10-31

**Soundness:** 3
**Presentation:** 2
**Contribution:** 2
**Rating:** 4
**Confidence:** 3

**Summary:**

This paper proposes ISER (Isolation-based Spherical Ensemble Representations), a new unsupervised anomaly detection method designed to overcome the limitations of existing approaches such as conflicting distribution assumptions, inefficiency, and poor handling of diverse anomaly types. ISER extends isolation-based methods by introducing hypersphere radii as proxies for local density, maintaining linear time and constant space complexity. A similarity-based scoring function compares ensemble representations with a theoretical anomaly reference pattern, enabling more consistent anomaly identification. The method also enhances the Isolation Forest framework to mitigate axis-parallel bias and improve local anomaly detection. Experiments on 22 real-world datasets show that ISER achieves superior performance over 11 baselines, demonstrating both efficiency and robustness across multiple domains.

**Strengths:**

1. Extensive experiments were conducted to analyze ISER across a wide range of datasets, with comprehensive comparisons against numerous baseline methods.
2. The proposed method demonstrates high computational efficiency and good scalability.

**Weaknesses:**

1. The motivation behind the proposed method is not clearly articulated. Although the paper introduces minor modifications to iNNE and IDK to address their respective challenges, the rationale for combining the two approaches remains unclear.
2. The method design relies heavily on empirical estimations and lacks solid theoretical justification for its key assumptions.
3. The paper’s overall exposition lacks coherence and contains several obvious errors and imprecise statements, giving the impression of being hastily written.

**Questions:**

1. The key assumption of the proposed method that the radius of a hypersphere can serve as a proxy for the local density of a sample point is questionable. Intuitively, local density should be reflected by the ratio of the number of sample points inside the hypersphere to its volume. Since the paper directly uses the hypersphere radius as a proxy for density, a stronger theoretical justification is necessary.
2. In Equation (4), the choice of the function $\frac{1}{r[\hat{z}_i(x)]}$ lacks explanation. Could other functions that are monotonically decreasing with respect to $r[\hat{z}_i(x)]$ and take values within $[0,1]$ be used instead? Would such alternatives affect the method’s performance, and should a comparative analysis be conducted?
3. The proposed method uses similarity or mean-based strategies to compute anomaly scores. Although this reduces computational cost, both strategies are based on empirical observations rather than rigorous theoretical guarantees like those provided in IDK [2].
4. In Figure 1, the notation $Di$ is clearly incorrect, as the index $i$ is not shown as a subscript. Moreover, the bold points in the figure, which seem to represent sampled points, are not explicitly explained, and there is no legend provided. Is it necessary to make these additions?
5. In Table 5, the %Ano value for the “fault” dataset is clearly wrong, which raises concerns about the reliability of the experimental code. Additionally, in the authors’ anonymous repository, the “Avg.” row in the README’s version of Table 5 is inconsistent with the version presented in the paper.
6. The abstract mentions the “difficulty in handling different anomaly types,” yet the introduction contains no discussion of this issue. This inconsistency should be addressed.
7. The introduction does not make the logic behind combining the two methods clear. It only shows small modifications made to iNNE [1] and IDK [2] to overcome their respective challenges, but why should these two methods be combined at all?

[1] Bandaragoda, Tharindu R., et al. "Isolation‐based anomaly detection using nearest‐neighbor ensembles." _Computational Intelligence_ 34.4 (2018): 968-998.
[2] Ting, Kai Ming, et al. "Isolation distributional kernel: A new tool for kernel based anomaly detection." _Proceedings of the 26th ACM SIGKDD international conference on knowledge discovery & data mining_. 2020.

---

> ### Author Response · Authors · 2025-11-21
>
> We thank the reviewer for the careful reading and constructive feedback. We address them as follows:
>
> **1. Motivation and method design**
>
> ISER does **not** combine iNNE and IDK, it uses the same hypersphere partitioning mechanism but introduces fundamentally different encoding and scoring. Appendix K formalizes why ISER's design overcomes limitations of both predecessors. The motivation is to address existing isolation-based anomaly detection methods' known limitations through principled geometric encoding.
>
> **2. Theoretical justification for radius as density proxy**
>
> We provide proof in newly added **Appendix C** that under random sampling, the expected hypersphere radius satisfies $\mathbb{E}[r] = C \cdot \rho^{-1/d}$ through geometric analysis, which means that hypersphere radii can be used as a monotonic transformation of local density. The derivation shows that while radius does not linearly approximate density, it provides a monotonic transformation preserving density ordering, where density-based anomaly detection methods consider that the point with higher local density is more likely to be an anomaly.
>
>
> **3. Choice of $1 - \frac{1}{r[\hat{z}_i(x)]}$:**
>
> Any strictly monotonic function of $r$ would preserve density ordering (Appendix C). We chose $1 - \frac{1}{r[\hat{z}_i(x)]}$ because it produces an upper range where $\phi \to 1$ for sparse regions or outside of all hyperspheres (large $r$), and $\phi \to -\infty$ for dense regions (small $r$), facilitating the similarity-based scoring in ISER-S by providing a clear reference anomaly vector (all ones vector $[1, \dots, 1]$).  Alternative monotonic functions would theoretically work but require different reference values for similarity computation.
>
> **4. Theoretical guarantees for scoring strategies**
>
> Appendix C now provides theoretical foundations: the monotonic relationship $r \propto \rho^{-1/d}$ guarantees that both average-based and similarity-based scoring preserve density-based anomaly ordering. While IDK provides kernel-theoretic guarantees, ISER offers geometric guarantees. The analysis of two scoring methods are provided in Apendix J.
>
> **5. Notation and presentation errors**
>
> We thank the reviewer for identifying these issues:
> - **Figure 1**: We have corrected the caption of Figure 1.
> - **Table 5 fault dataset information**: The %Ano value is indeed incorrect (should be 34.67%). This was a data entry error that does not affect experimental results as all methods used the correct dataset. We have fixed this in revision.
> - **README inconsistency**: The result in the paper submission version is correct, we have already corrected in the repository.
> - **Abstract-Introduction mismatch**: We have added description of different anomaly types to the introduction for consistency.
>
> Thank you again for your valuable comment.

---

### Official Review · Reviewer_1PRx · 2025-10-31

**Soundness:** 2
**Presentation:** 2
**Contribution:** 2
**Rating:** 2
**Confidence:** 4

**Summary:**

The paper introduces isolation-based spherical ensemble representations which extends isolation-based anomaly detection through hypersphere-based partitioning and ensemble scoring. The approach aims to approximate local density using hypersphere partioning while. The authors report performance improvements over several baselines on synthetic and real-world datasets. The work is clearly written and demonstrates the proposed approaches in experimental evaluation. However, several conceptual and analytical aspects remain underdeveloped, particularly regarding theoretical grounding and the interpretation of experimental results.

**Strengths:**

- The paper provides an accessible overview of existing isolation-based methods and motivates the need for improving density awareness.
- ISER maintains the desirable linear-time and constant-space properties of traditional isolation methods. The scalability analysis is presented and demonstrates practical efficiency.
- Code for reproducing the results is provided.

**Weaknesses:**

- Incremental novelty. Hypersphere partitioning-based isolation approach was used in iNNE and IDK.
- Lack of formal theoretical grounding. The definitions of global, local, and dependency anomalies rely on qualitative descriptions such as "deviate significantly" or "sparse regions," without mathematical/statistical formulations. Consequently, it is unclear how the proposed mechanism theoretically distinguishes among these anomaly types. The role of hypersphere radii as density proxies is intuitive but not formally justified.
- Incremental perofrmance improvement. Figure 2 aggregates results from heterogeneous datasets with different baseline performances, making it challenging to interpret relative improvement. Figure 3 shows that ISER is statistically comparable to several existing methods, suggesting only marginal differences. While visually clear, these figures provide limited insight into mechanism or consistency.
- Moreover, the authors provide limited analysis of why the method performs better or under what data conditions it is expected to succeed.
- Although ISER has linear-time complexity wrt $n$, this advantage is shared with existing methods. The reported runtime gains are minor and largely reflect constant-factor efficiency improvements rather than new theoretical insights into computational scalability.

**Questions:**

- The proposed method does not seem to involve representation learning in the usual sense but rather constructs a fixed mapping from random hypersphere partitions but more like a randomized geometric feature transformation. Since the mapping $\Phi$ is not learned, I wonder if this fits within unsupervised representation learning. Please correct me if I’m mistaken.
- The authors do not provide any results/analyses about how the hypersphere partitioning behaves in high-dimensional settings, where nearest neighbor become less informative. It is unclear whether the proposed method remains meaningful as the dimension increases.
- It is unclear how the hyperparameters were selected. If I understood correctly, this work focuses on the unsupervised anomaly detection and there's no (known) anomalies in the validation set.

---

> ### Author Response · Authors · 2025-11-21
>
> We thank the reviewer for the detailed assessment. We address each concern as follows:
>
> **W1: Incremental novelty over iNNE/IDK**
>
> While hypersphere partitioning is shared, ISER introduces fundamentally different encoding and scoring mechanisms. Newly added Appendix C formalizes this distinction: ISER uses absolute radius encoding $\phi = 1 - 1/r$ where we prove $\mathbb{E}[r] \propto \rho^{-1/d}$, avoiding iNNE's ratio-based approach that fails when neighboring hyperspheres have similar radii, and IDK's point-count embedding that fails when different densities have equal point counts (Appendix K). Additionally, ISER-S introduces similarity-based scoring against a theoretical anomaly reference pattern, which is conceptually distinct from prior methods. The contribution lies not in the partitioning mechanism but in how geometric properties are encoded and interpreted.
>
> **W2: Lack of theoretical grounding**
>
> We add Appendix C to provide:
> - Proof that hypersphere radii satisfy $\mathbb{E}[r] = C \cdot \rho^{-1/d}$ via geometric derivation from hypersphere volume analysis.
> - Proof that the transformation $\phi_i(x) = 1 - \frac{1}{r[\hat{z}_i(x)] }$ preserves a monotonic transformation of local density.
>
> Regarding anomaly type definitions, we adopted the standard definition commonly used in anomaly detection research (Han et al. (2022)). The mechanism's ability to distinguish these types is validated empirically in Table 1 and discussed theoretically in Appendix C for local anomalies.
>
> **W3: Performance improvement appears incremental**
>
> Figure 3 demonstrates that ISER-S is statistically significantly better than 9 of 11 baselines (Friedman-Nemenyi test), with only iNNE and IDK showing no significant difference, but ISER still gets better average performance than them. It indicates consistent superiority across diverse datasets.
>
> **W4: Analysis of why the method performs better:**
>
> The method succeeds when datasets exhibit varying local densities (a common situation in most datasets), exactly the scenario where density-aware detection is needed. ISER is able to detect global, local and dependency anomaly types. Performance across 22 datasets with diverse characteristics (varying dimensions and anomaly rates) demonstrates consistent superiority. While no method universally dominates all datasets due to inherent data heterogeneity, ISER's theoretical guarantees and linear complexity provide advantages in tabular anomaly detection scenarios. Appendix K explains how ISER addresses the limitations of iNNE and IDK.
>
> **W5: Runtime advantages are minor**
>
> The comparison in Figure 4 demonstrates that ISER methods are **faster** than IDK (the most competitive baseline) by substantial margins. ISER achieves approximately 3-5× speedup over IDK across varying sample sizes. This is not a theoretical claim but an empirical demonstration of practical efficiency gains from avoiding IDK's expensive kernel mean embedding computations.
>
> **Q1: Is this representation learning?**
>
> ISER performs geometric feature transformation rather than learned representation in the neural network sense. We do not claim to perform representation learning, the term "ensemble representations" in our title refers to ensemble feature vectors $\Phi(x) \in \mathbb{R}^t$ constructed through geometric encoding. This approach falls within the paradigm of classical unsupervised learning, as it constructs features without any label supervision.
>
> **Q2: High-dimensional behavior**
>
> Our experimental evaluation includes datasets up to 500 dimensions (fault dataset), with ISER demonstrating best performance in terms of AUROC. The concern about nearest neighbors becoming less informative in high dimensions is valid for distance-based methods, but ISER's ensemble mechanism provides robustness: aggregating over multiple independent random partitionings mitigates the instability of individual nearest-neighbor computations.
>
> **Q3: Hyperparameter selection**
>
> ISER has two hyperparameter, number of random samples $\psi$ and number of partitionings $t$.As stated in **Appendix E**, we perform grid search over hyperparameters: $\psi \in \{2, 4, 8, ..., 256\}$ and t=200 (default). **Appendix I** provides the hyperparameter sensitivity analysis showing ISER is sensitive to number of sample points $\psi$ but not sensitive to number of partitionings $t$. In the unsupervised anomaly detection setting, the test set contains no labeled anomalies by definition, the problem formulation is formally described in Section 2.
>
> Thank you again for your valuable comment.

---

### Official Review · Reviewer_kePE · 2025-11-01

**Soundness:** 2
**Presentation:** 2
**Contribution:** 2
**Rating:** 4
**Confidence:** 3

**Summary:**

This work studies on the problem of anomaly detection and proposes a new method based on isolation-driven spherical ensemble representations.

While the topic is important and the paper presents a method that is conceptually reasonable, I have significant concerns on the motivation and the experimental evaluation. As a result, I assign a score of 4.

Notably, I am not familiar with this research area.  While I have spend considerable time on reviewing this work, my comments may lack field-specific depth.

**Strengths:**

1.	This work studies on an important problem.
2.	The proposed method appears to be reasonable.
3.	Experiments on 22 datasets have been conducted to verify the efficacy of the proposed method.

**Weaknesses:**

1.	My major concern lies on the unclear motivation. Some important questions have not been addressed, like: 1) deep-learning-based anomaly detection methods have been widely explored by recent work, and have demonstrated high effectiveness. The paper’s stated drawbacks --- such as longer training time and reduced interpretability --- are not convincingly shown to be critical limitations in practice.  The authors should discuss the actual severity of these issues and their impact on real-world deployments. 2) The introduction of hypersphere radii is interesting, but the underlying intuition and its specific benefits for the proposed method remain unclear. More discussions should be provided.

2.	The proposed method is largely heuristic. Note that the proposed method is relatively traditional. Without a solid theoretical justification and analysis, the technical contributions appear limited.


3.	There are also some significant limitations on the experiments: 1) The baselines are relatively old (all before 2023). More state-of-the-art methods (e.g., [a1]) should be included.  2) From the table 7, the improvements of the proposed method over the best baseline are modest, and the proposed method does not achieve state-of-the-art performance on many datasets. This raises questions about the practical significance of the approach.

4.	Some concerns on the presentation: 1) Many important contents (e.g., the experimental setup and comparable results) are presented in the appendix, making them less accessible to readers. 2) Tables 6 and 7 are formatted horizontally, which heavily affects readability.

[a1] cvpr’25: Dinomaly: The less is more philosophy in multi-class unsupervised anomaly detection

**Questions:**

Please refer to weaknesses.

---

> ### Author Response · Authors · 2025-11-21
>
> We thank the reviewer for the constructive feedback. We provide responses to each of your comments as follows:
>
> **W1: Research motivation**
>
> As stated in Section 2, we address unsupervised anomaly detection where datasets contain unlabeled mixtures of normal and anomalous instances. Most deep learning methods (e.g., AutoEncoder) require large amounts of clean normal data for training to learn normal pattern, which is rarely available in real-world scenarios where anomalies are unlabeled and intermixed with normal instances. Additionally, deep methods suffer from high computational complexity during iterative training and substantial memory overhead. ISER directly tackles the unsupervised setting with linear time complexity $O(n\psi t)$ and constant space complexity $O(\psi t)$.
>
> **W2: Theoretical foundation for hypersphere radii**
>
> The newly added **Appendix C** provides theoretical justification for using hypersphere radii as density indicators. We prove that under multiple random sampling, the expected radius satisfies $\mathbb{E}[r] = C \cdot \rho^{-1/d}$ through geometric analysis of hypersphere volumes. This establishes a formal monotonic relationship between radii and local density.
>
>
> **W3: Experimental limitations**
>
> We first want to clarify ISER targets unsupervised **tabular** anomaly detection, which is fundamentally different problem domains and settings (Dinomaly (CVPR'25) addresses **image** anomaly detection). Our baseline selection covers the major algorithmic families for unsupervised tabular anomaly detection, including recent deep methods (DTE, ICLR 2024). We acknowledge that no single method achieves universal superiority across all datasets, which is expected given dataset heterogeneity. However, ISER achieves competitive performance with best average rankings and significant better than 9/11 methods, its primary advantages lie in computational efficiency, linear time and constant space complexity enable scalable deployment on large datasets where quadratic complexity methods become infeasible.
>
> **W4: Presentation**
>
> We have improved accessibility by adding explicit forward references to appendices throughout the main text and reformatting Tables 6-7 vertically for better readability.
>
> Thank you again for your valuable comment.

---

### Official Review · Reviewer_vDkB · 2025-11-01

**Soundness:** 3
**Presentation:** 3
**Contribution:** 2
**Rating:** 4
**Confidence:** 4

**Summary:**

This paper introduces ISER, a novel unsupervised anomaly detection algorithm. The method builds upon existing hypersphere-based partitioning techniques (like those in iNNE and IDK) but introduces a new way to encode and score data points. The core of ISER is to construct an ensemble representation for each data point. This is done by repeatedly sampling subsets of data, defining hyperspheres based on nearest-neighbor distances within these subsets, and then encoding based on its nearest hypersphere. The paper proposes two scoring mechanisms for this representation, ISER-A and ISER-S. Furthermore, the paper proposes ISER-IF, which uses the ISER ensemble representation as a transformed feature space for the standard iForest. The authors correctly identify that this transformation inverts iForest's core assumption and propose a modified scoring function to address iForest's axis-parallel bias.

**Strengths:**

1. The paper provides a clear theoretical justification for why this absolute density proxy is more robust than iNNE's relative (ratio-based) score and IDK's point-count-based embedding. This new representation successfully avoids the failure modes of its predecessors.
2. The paper introduces two scoring methods. The similarity-based ISER-S is particularly innovative.
3. The ISER-IF contribution is also a major strength.
4. The paper is backed by a large-scale, comprehensive empirical study. The use of 22 real-world datasets + 3 synthetic datasets against 11 baselines provides strong evidence.

**Weaknesses:**

1. The paper claims that it follows the same hypersphere partitioning as iNNE/IDK, and then contrasts scoring qualitatively without providing formal conditions where ISER is better.
2. Assigning a flat score of 1 to all points outside their nearest hypersphere (Eq. 4) is a questionable design, as this binary assignment discards information about the degree of isolation. The authors should provide a justification for this.
3. The paper claims that normal data and anomalies cluster in $\Psi$-space, leading to longer anomaly path lengths and a flipped scoring rule. This is plausible but needs more clarifications. Please supply theory or empirical evidence showing when this ordering holds.
4. The paper states that points in overlapping regions are assigned only to the nearest hypersphere center. Is this strategy still valid when a point lie at the junction of a "sparse" large sphere and a "dense" small sphere.
5. The core design choices (e.g., the $1 - 1/r$ form for $\phi_i(x)$) are justified by intuition rather than formal arguments. The authors should provide theoretical analysis to support their method.

**Questions:**

Please refer to Weaknesses.

---

> ### Author Response · Authors · 2025-11-21
>
> We thank the reviewer for the constructive feedback. We provide detailed responses to each of your comments as follows:
>
> **W1: Formal conditions for ISER superiority**
>
> Refer to Appendix K: iNNE's ratio-based score $1-\frac{r[z]}{r[\eta[z]]}$ fails when neighboring hyperspheres have similar radii (ratio $\approx$ 1) although they are in sparse regions, losing discriminative power regardless of actual density. ISER avoids this by using encoding $\phi=1-1/r$, which maintains separation as long as **local densities differ**, since $r \propto \rho^{-1/d}$ (proven in new Appendix C).
>
> **W2: Why assign $\phi=1$ to uncovered points**
>
> Points falling outside all hyperspheres are more likely to be **local anomalies**, assigning $\phi=1$ gives these uncovered points the maximum anomaly indicator within each partitioning. Across the ensemble, local anomalies exhibit $\phi=1$ more frequently than normal points, enabling discrimination through aggregated scores. This mechanism also provides a consistent upper bound for similarity-based scoring in ISER-S. The effectiveness is validated in Table 1.
>
> **W3: Evidence for ISER-IF feature space inversion**
>
> The newly added Figure [6] demonstrates the feature space transformation. In the original space (left), anomalies are scattered while normal points cluster. In $\Phi$-space (right), this inverts: anomalies cluster tightly, normal points scatter. The reason is, anomalies in sparse regions either fall outside hyperspheres ($\phi=1$) or within large-radius hyperspheres ($\phi \approx 1$). Both yield high $\phi$ values across partitionings, producing concentrated $\Phi$-representations. Normal points in dense regions fall within hyperspheres of diverse small radii, producing varied $\Phi$ values and scattered $\Phi$-representations. This inversion justifies the modified scoring in Eq. (9): in $\Phi$-space, the anomalies tend to be a cluster and "hard to isolate" (longer paths in iForest applied to $\Phi$), necessitating the inverted scoring rule to correctly identify them as anomalies.
>
>
>
> **W4: Boundary points between dense/sparse regions**
>
> Single partitionings may assign boundary points to either dense or sparse hyperspheres. However, across multiple independent partitionings, boundary points naturally receive mixed assignments, sometimes from dense regions (low $\phi$), sometimes from sparse regions (high $\phi$). Ensemble aggregation produces intermediate scores reflecting their transitional nature. This is evidenced by smooth decision boundaries in Table 1 at density transitions.
>
> **W5: Theoretical foundation for core design**
>
> We now provide mathematical justification in newly added Appendix C:
>   - Random sampling preserves relative density (sampling density ratio = original density ratio).
>   - Hypersphere radius satisfies $\mathbb{E}[r] = C \cdot \rho^{-1/d}$ via geometric derivation from hypersphere volume formula.
>   - Transformation $\phi_i(x) = 1 - \frac{1}{r[\hat{z}_i(x)] }$ preserves density ordering (monotonic relationship).
>
> - **Key insight**: Density-based anomaly detection assumes anomalies reside in lower local density regions, requiring only preservation of relative density ordering rather than absolute density values. The proven monotonic relationship $r \propto \rho^{-1/d}$ is both necessary and sufficient.

---

### Meta-Review · Area_Chair_sKKt · 2026-01-05

**Summary:**

The paper proposes ISER (Isolation-based Spherical Ensemble Representations), an unsupervised tabular anomaly detection method that leverages hypersphere radii as a proxy for local density. The method aims to improve upon existing isolation-based methods (like iForest, INNE, IDK) by maintaining linear time complexity while addressing issues like axis-parallel bias.

**Reviewer Concerns:**

Reviewers generally appreciated the computational efficiency and the extensive empirical evaluation across 22 datasets. However, the initial consensus was negative (scores: 2, 4, 4, 4), driven by several shared concerns:

Theoretical Grounding: Multiple reviewers (vDkB, kePE, 1PRx, Kd7K) questioned the theoretical justification for using hypersphere radii as a direct proxy for density and the specific transformation function used.

Incremental Novelty: Reviewers (1PRx, Kd7K) felt the method was a minor modification or combination of existing methods (INNE/IDK) rather than a distinct contribution.

Motivation & Baselines: Concerns were raised regarding the motivation vis-à-vis deep learning methods and the age of the baselines selected (kePE).

Performance Margins: Reviewers noted that improvements over baselines appeared marginal or difficult to interpret (1PRx, kePE).

**Reviewer Scores:**

While the baseline justification (tabular vs. image) is valid, reviewer kePE concern about the method being "traditional" and "heuristic" might prevent a stronger endorsement. The theoretical additions address concerns of reviewer 1PRx  about "lack of formal grounding," but he/she viewed the novelty as fundamentally incremental, a view that often persists despite clarification on encoding differences.

---

### Decision · Program_Chairs · 2026-01-26

Reject